# Volatile Profile Characterization of Jujube Fruit via HS-SPME-GC/MS and Sensory Evaluation

**DOI:** 10.3390/plants13111517

**Published:** 2024-05-31

**Authors:** Ruojin Liu, Ling Ma, Xiangyu Meng, Shuwei Zhang, Ming Cao, Decang Kong, Xuexun Chen, Zhiqin Li, Xiaoming Pang, Wenhao Bo

**Affiliations:** 1State Key Laboratory of Tree Genetics and Breeding, National Engineering Research Center of Tree Breeding and Ecological Restoration, Key Laboratory of Genetics and Breeding in Forest Trees and Ornamental Plants, Ministry of Education, College of Biological Sciences and Biotechnology, Beijing Forestry University, Beijing 100083, China; 13244000808@163.com (R.L.); lingmaml@163.com (L.M.); mengxiangyu1302@163.com (X.M.); zsw2000@bjfu.edu.cn (S.Z.); xmpang@163.com (X.P.); 2National Foundation for Improved Cultivars of Chinese Jujube, Cangzhou 061000, China; i2008caoming@126.com (M.C.); kongdecang@126.com (D.K.); 3Bureau of Forestry of Aohan, Chifeng 028000, China; xuexunchen@163.com; 4Agricultural Comprehensive Service Center, Dong Lianhuayuan Town, Qianxi County, Tangshan 063000, China; hydz5982012@126.com

**Keywords:** jujube, volatile compounds, glycoside bound compounds, sensory evaluation

## Abstract

Current research does not fully elucidate the key compounds and their mechanisms that define the aroma profile of fresh jujube fruits. Therefore, this study conducted a comprehensive analysis of both free and glycosidically bound aroma compounds in fresh jujube fruits of ten cultivars. Utilizing gas chromatography–mass spectrometry (GC-MS), we identified 76 volatile free aroma compounds and 19 glycosidically bound volatile compounds, with esters, aldehydes, and ketones emerging as the predominant volatile compounds in the jujube fruits. Odor activity value (OAV) analysis revealed that the primary aroma profile of the jujubes is characterized by fruity and fatty odors, with β-damascenone being a key contributor to the fruity aroma, and (E)-2-oct-en-1-al and nonanal significantly influencing the fatty aroma. Moreover, the integration of sensory evaluation and partial least squares regression (PLSR) analysis pinpointed octanal, (E)-2-oct-en-1-al, nonanal, β-damascenone, and pentanal as significant contributors to the jujube’s characteristic aroma, while isoamyl acetate was identified as significantly influencing the fatty acid taste. This study not only underscores the complexity of the jujube aroma composition but also highlights the impact of environmental factors on aroma profiles, offering valuable insights into the sensory characteristics of jujube fruits.

## 1. Introduction

Jujubes (*Ziziphus jujuba* Mill.) are native to China, where they have been cultivated for more than 4000 years [1]. Fresh jujube is not particularly juicy, but the taste is delicious, the appearance is slippery and crisp, and the pulp is sweet but not greasy [2]. It is one of the most popular high-end fresh fruits born from deciduous fruit trees in northern China [3]. Jujube is rich in fructose, dietary fiber, organic acids, phenolics, polysaccharides, vitamins, and trace elements required by the human body, providing a variety of nutrition and benefits to the health of consumers [4,5]. It has been reported that jujube high contents of fructose and fiber may contribute to control calorie intake and regulate blood sugar, and other bioactive contents in jujube have anti-inflammatory, antioxidant, anticancer, and anti-obesity effects [6,7]. As a health food, jujube is often used as a natural Chinese medicine, having the function of strengthening the body, soothing coughing, and improving immunity [8]. Due to its rich nutritional value, it has become one of the more popular medicinal and edible fruits on the Chinese market.

Aroma is one of the most valued factors of jujube and can also bring sensory pleasure to the consumer for eating [9]. There are many factors that influence jujube aroma, including biological variety (cultivar), climate (especially warmth and rainfall), geographical location, soil geology, and agricultural practices (like time of harvest and storing/drying methods) [10]. This multifaceted influence underscores aroma’s critical role in the fruit’s appeal, offering sensory pleasure and enhancing the eating experience [9]. Such insights highlight the complexity and importance of understanding the determinants of jujube aroma for both consumer satisfaction and agricultural practices. The volatile substances in fruits are composed of free aroma and glycoside-bound aroma [11]. Recent studies have shed light on the intricate composition of volatile components in jujube fruits, revealing a rich palette of aroma compounds that vary with maturity stages, drying methods, and geographical origins. 

Song et al. (2019) conducted an analysis of the volatile components in jujube fruits at various stages of maturity, discovering that the primary free aroma compounds in jujubes are acids, aldehydes, and esters [12]. Chen et al. (2018) utilized HS-SPME/GC-MS and electronic nose technology to analyze the volatile components in ten different cultivars of fresh jujubes and identified 51 types of free aroma compounds, in which aldehyde compounds are the main contributors to the jujube’s aroma [13]. Song et al. (2022) applied HS-SPME/GC-MS and electronic nose technology to characterize the aroma of dried jujubes, identifying acetic acid, hexanoic acid, butanoic acid, and 2-methylbutanoic acid as the primary aroma components in both red jujube and heat pump dried jujube [14]. Liu et al. (2021) employed HS-SPME/GC-MS and an electronic nose to analyze the volatile characteristics of five Xinjiang jujube cultivars, discovering that acids, aldehydes, and esters are the main aroma components in these Xinjiang jujube cultivars [15]. Qiao et al. (2021) utilized HS-SPME-GC/MS and GC-IMS for a combined analysis to characterize the aroma of winter jujubes from different origins, finding significant differences in the aroma profiles of winter jujubes from various locations [16]. Yan et al. (2023) used microwave drying technology to identify the aroma-active substances in dried jujube slices [17]. Through sensory evaluation, they discovered that the slices’ characteristic aromas include caramel, roasted sweet, and bitter flavors, primarily determined by ketone and furan compounds.

Presently, the scientific literature reveals a substantial gap in the theoretical foundation concerning the pivotal compounds influencing the aroma profile of fresh jujube fruits. Notably, discussions on glycosidically bound aroma constituents within fresh jujubes remain conspicuously sparse. Glycosidically bound compounds exist as potential odorants that require specific biochemical reactions to release the actual aroma substances, known as aglycones. Consequently, this research endeavors to elucidate the potential impact of the glycosidically bound components of fresh jujubes. In sensory evaluation, partial least squares regression (PLSR) effectively elucidates the relationships between extensive sensory data and target variables. This method holds potential for further application in the study of fresh jujube aromas. To this end, this study utilized headspace solid-phase microextraction gas chromatography–mass spectrometry (HS-SPME-GC-MS) for the comprehensive detection of both free and glycosidically bound compounds across various harvest years of fresh jujube cultivars. Furthermore, the study quantitatively evaluated the contribution of free aroma compounds to the jujube’s primary characteristic aroma through the calculation of the odor activity value (OAV). In culmination, through sensory evaluation and partial least squares regression (PLSR) analysis, the study clarified the correlation between sensory characteristics and the presence of characteristic aromatic active compounds.

## 2. Results and Discussion

### 2.1. Free and Bound Volatile Compositions

In this study, both free aroma compounds and glycosidically bound compounds of fresh jujube were carefully analyzed using GC-MS, with the volatile compounds catalogued in Table 1. Drawing on insights from previous research into the aroma components of fresh jujube, it is evident that alcohols, acids, aldehydes, and esters play pivotal roles in crafting the distinct aroma profiles of various jujube cultivars. Our comprehensive analysis revealed a total of 86 free and bound substances within our fresh jujube samples. Among these, we identified 76 free substances, including 9 acids, 5 alcohols, 14 aldehydes, 3 phenols, 12 esters, 10 ketones, 10 terpenoids and isoprenoids, 10 benzenoid aromatics, and 3 miscellaneous substances (Table 2). Furthermore, we detected 19 bound volatiles, encompassing 6 acids, 10 alcohols, 1 phenol, and 2 terpenoids, enriching our understanding of the complex aroma profile of fresh jujube (Table 3).

#### 2.1.1. Esters

Esters have been recognized for their significant contribution to the development of fruity, floral, and sweet aromas that enhance the overall scent profile of fruits [18]. This investigation revealed a varied ester content across ten distinct fresh jujube cultivars. Notably, in 2020, the ‘TPJDZ’ cultivar demonstrated the most substantial presence of free esters, reaching 10,025.65 μg/L, in stark contrast to the ‘YCXZ’ cultivar, which recorded the minimal concentration at 1239.36 μg/L. Contrary to fruits like apples, pineapples, and cherries, known for their rich ester aromas, these jujube cultivars were characterized by comparatively lower levels of esters.

In this study, we detected 12 free volatile esters, including methyl acetate, ethyl acetate, ethyl propanoate, ethyl 2-methylbutanoate, isoamyl acetate, ethyl hexanoate, hexyl acetate, ethyl heptanoate, ethyl lactate, ethyl 2-hexenoate, ethyl caprate, and ethyl dodecanoate. Notably, ethyl acetate stood out for its fruity and sweet aroma, becoming the most prevalent free ester in these cultivars with a peak concentration of 9775.09 μg/L. Consistent with findings by Yang et al. (2019), ethyl acetate, along with ethyl propanoate and hexyl acetate, were identified as key volatile components, especially significant at the end of storage, and were also present in our findings [19]. Highlighted by Ayala-Zavala et al. (2005), Neri et al. (2015) and Qin et al. (2014), methyl acetate was recognized for its sweet, fruity flavors reminiscent of apples and grapes, commonly found in strawberries and pears [20,21,22]. Our analysis revealed that the levels of free ethyl acetate, ethyl 2-methylbutanoate, and ethyl hexanoate surpassed their threshold levels, underscoring their vital roles in defining the fresh jujube’s aroma profile. In particular, free ethyl acetate demonstrated an odor activity value (OAV) greater than 1 in the cultivars ‘JSBZ’, ‘TPJDZ’, ‘XZHZ’, ‘BXJZ’, and ‘CXJSXZ’ in 2020. Ethyl hexanoate, known for its apple, peel, and fruity aromas, was found in concentrations exceeding its threshold across all tested jujube cultivars, significantly contributing to the overall aroma. Ethyl 2-methylbutanoate, with its distinctive sweet, green, apple, and fruity notes and a low flavor threshold (0.006 μg/L), significantly enhanced the flavor of fresh jujube, with notable OAV values between 1628.33 and 2358.33 in the ‘BXJZ’ and ‘ZYDZ’ cultivars for both 2020 and 2021. However, our study did not detect any bound esters in the jujube cultivars.

#### 2.1.2. Aldehyde and Ketones

The GC-MS results of this experiment indicate that aldehydes and ketones also occupy an important position among the volatile substances in fresh jujube. Francisca Hernández (2015) and Qin et al. (2017) also used GC-MS to detect aldehyde and ketone aroma compounds in fresh jujube [13,23]. In this study, we detected a total of 15 free and glycosidically bound aldehyde compounds, underscoring their significance within the volatile profile of the sample. Notably, several of these compounds, including hexanal, nonanal, and benzaldehyde, have been previously identified in the literature, reflecting a consistency with previous studies [24,25]. In this study, the GC-MS results revealed the presence of free aldehydes across all cultivars, including isopentanal, pentanal, (E)-but-2-en-1-al, hexanal, heptanal, (E)-hex-2-en-1-al, octanal, (Z)-hept-2-en-1-al, nonanal, (E)-2-oct-en-1-al, decanal, non-2-en-1-al, (E)-dec-2-en-1-al, benzaldehyde, and cumaldehyde. Consistent with the existing literature, benzaldehyde has been identified as a significant volatile component in jujube [26].

The benzaldehyde content in various fresh jujube cultivars ranged from 6.67 μg/L to 586.44 μg/L, characterized by its sweet and cherry flavor notes [24]. This content varied significantly across different cultivars. However, the concentration of benzaldehyde did not surpass the detection threshold, thereby constraining its influence on the fresh jujube’s aroma profile. Hexanal is also an important aldehyde compound, which is consistent with previous studies [13,23,24]. The concentration of hexanal in fresh jujube varied from 26.00 μg/L to 263.13 μg/L, imparting a distinctive grass-like and green fragrance. Its odor activity value exceeds 1, indicating that its green flavor plays a significant role in enhancing the overall aroma of fresh jujube. While our research revealed relatively low concentrations of (E)-hex-2-en-1-al, ranging from 7.49 μg/L to 19.17 μg/L, prior studies have documented significantly higher levels, with concentrations spanning from 534.00 μg/L to 1748.30 μg/L [13]. This variance could be attributed to differences in jujube cultivars and their geographical origins.

Isopentanal and pentanal play significant roles in the aroma profile of fresh jujube. Pentanal, characterized by its fruity, nutty, and berry-like aromas, although prevalent in dried jujube and jujube (*Ziziphus jujuba* miller) jams, had not been previously detected in the aroma of fresh jujube [7,27,28]. In this investigation, we have successfully quantified the concentrations of pentanal in fresh jujube, elucidating its significant contribution to the fruit’s aroma profile. In 2020, pentanal levels ranged from 39.06 μg/L to 187.31 μg/L and were detected across all cultivars, with concentrations exceeding the detection threshold. Additionally, pentanal is also a common constituent in various red fruits, such as cherry tomatoes, grapes, and mulberries [29,30], indicating its widespread occurrence in fruit aromatics. Isopentanal, distinguished by its rich chocolate, peach, and fatty aromatics, serves as a pivotal flavor compound, contributing significantly to the distinctive aroma profiles of tomatoes, both fermented (including green or oxidized beer and raw spirits) and non-fermented (heat-treated) products [7,31,32,33]. In this study, the detected concentrations of isopentanal significantly exceeded the olfactory threshold for both 2020 and 2021. Specifically, in 2020, isopentanal levels varied from 15.58 μg/L to 312.72 μg/L. In the subsequent year, 2021, the concentration range of isopentanal was observed between 59.32 μg/L and 275.49 μg/L.

Despite their low concentrations, specific aldehydes significantly enhance the aromatic profile owing to their minimal olfactory detection thresholds, each imparting distinct aroma characteristics. Octanal exhibits notes of citrus, orange, green, and fatty qualities, while nonanal is characterized by its rose and orange aromas. (E)-2-Oct-en-1-al is notable for its cucumber-like green, herbaceous, and fatty flavors, and non-2-en-1-al presents a combination of fatty, green, and cucumber notes. Moreover, (E)-dec-2-en-1-al is marked by its unique fatty, earthy, cilantro, green, and mushroom aromas, with decanal adding notes of oil, orange, and fruit peel. The presence of these compounds at concentrations surpassing their sensory thresholds underscores their critical role in defining the unique aroma and flavor profile of fresh jujubes. Parallel findings have been reported in other studies focusing on jujube dates, where similar volatile compounds were detected [9,15,19].

In our research, we detected eleven distinct free ketones in fresh jujube fruits, whereas Yang et al. revealed forty-seven aromatic compounds in jujube trees, including six ketones, using headspace gas chromatography–ion mobility spectrometry (HS-GC-IMS) [9]. Aligning with previous research, we also detected pentan-3-one and nonan-2-one, which was previously reported in the literature [9,13]. Notably, our investigation unveiled eight ketones in fresh jujube cultivars for the first time, i.e., hex-4-en-3-one, pent-3-en-2-one, isobutyl ketone, heptan-2-one, acetoin, 1-hepten-3-one, 2-methyl-3-octanone, and octan-2-one. Among the identified compounds, the concentrations of pentan-3-one and isobutyl ketone exceeded the olfactory detection threshold, markedly enriching the aromatic profile of the jujubes. Moreover, the free isobutyl ketone demonstrated an odor activity value greater than 1 across all cultivars in both 2020 and 2021, contributing to the distinctive aroma of fresh jujubes with its green, fruity, pineapple, and banana flavor characteristics.

#### 2.1.3. Acids

We quantified a total of 10 acids, both free and glycosidally bound, across various jujube cultivars. Regarding individual free acids, these cultivars contained acetic acid, butyric acid, isovaleric acid, valeric acid, hexanoic acid, octanoic acid, nonanoic acid, decanoic acid, and dodecanoic acid. Notably, acetic acid emerged as the most significant flavor-impact acid, contributing uniquely to the aroma profile of jujubes [10]. Known for its pungent, sour, and vinegar-like odors, acetic acid’s odor activity value is above 1. Other major acids in fresh jujube cultivars include hexanoic acid, decanoic acid, and octanoic acid, which share fatty-like flavor notes, aligning with findings from previous studies [13,26]. Interestingly, butyric acid, previously identified in dried jujubes, was detected in fresh jujubes for the first time in our study [28]. However, the concentration of these free acids in fresh jujubes did not exceed their threshold levels, with the exception of acetic acid. Additionally, six glycoside-bound acids were identified, including acetic acid, formic acid, hexanoic acid, octanoic acid, nonanoic acid, and decanoic acid. The concentration of formic acid varied from 213.85 μg/L to 1377.25 μg/L, existing solely in the glycoside-bound form in fresh jujubes.

#### 2.1.4. Alcohols

Previous studies have indicated that fresh jujubes contain various alcohols [13,19], yet the specific contribution of these free alcohol compounds to the distinctive flavor and overall aroma profile of fresh jujube merits additional investigation. In this study, we identified five free alcohols in jujube cultivars: ethyl alcohol, pent-1-en-3-ol, methyl-2-butan-1-ol, oct-1-en-3-ol, and 2-ethyl-1-hexanol. Notably, pent-1-en-3-ol and oct-1-en-3-ol, with their earthy, green, and fatty aromas and odor activity values (OAV) exceeding 1, were found to significantly contribute to the characteristic alcoholic fragrance of fresh jujube. Additionally, our research uncovered ten glycoside-bound alcohols, eight of which are exclusively found in their glycoside-bound state in fresh jujube, including 3-methyl-butan-2-ol, hexan-2-ol, 2,7-dimethyl-4-octanol, 4-methyl-2-heptanol, 5-methyl-2-heptanol, and dodecan-5-ol. Particularly, the concentration of 3-methyl-butan-2-ol reached up to 284.60 μg/L in 2021, situated within its threshold range of 250–300 μg/L. This compound, noted for its fruity aroma, stands out as one of the most significant contributors to the unique scent profile of jujube. Conversely, dodecan-1-ol exhibits a detrimental effect on the aroma potential of jujube due to its low threshold (0.0152–0.0533 μg/L) and its association with earthy, soapy, waxy, and fatty aromas. Meanwhile, oct-1-en-3-ol is believed to significantly enhance the jujube’s aroma, imparting mushroom, earthy, and green notes. In summary, alcohols play a pivotal role in defining the potential aroma of jujube.

#### 2.1.5. Benzenoid Aromatics

In our study, we detected 10 kinds of free benzenoid aromatics. For the first time, a wide variety of benzene compounds have been detected in fresh jujubes. A total of 10 free benzenoid aromatics including styrene, methyl benzoate, ethyl benzenecarboxylate, naphthalene, analgit, isobutyl benzoate, benzyl alcohol, ethyl benzenepropanoate, β-methylnaphthalene, and α-calacorene. Different cultivars showed different benzenoid aromatics compositions. For example, methyl benzoate that had its odor activity value above 1 only existed in cultivar ‘XZHZ’, indicating that its floral flavor feature might be incorporated into the overall aroma of the cultivar ‘XZHZ’. However, other free benzenoid aromatics had an odor activity value below 1.

#### 2.1.6. Terpenoids and Isoprenoids

Terpenoids, characterized by their sweet, fruity, floral, and rose notes, are crucial natural flavor compounds and represent some of the most significant volatile substances found in kiwifruit, citrus, and apple. This study detected nine types of terpenoids, including eucalyptol, sulcatone, camphor, linalool, α-ionene, hotrienol, levomenthol, α-terpineol, and cis-geranylacetone, as well as one C13 isoprenoid (beta-damascenone). Notably, of these compounds, β-damascenone emerged as the predominant one, renowned for its intense rose, berry, and sweet flavors, with concentrations ranging from 1.68 μg/L to 5.83 μg/L. The significant impact of β-damascenone on the aroma of fresh jujube is attributed to its low threshold (0.20–0.90 ng/mL) [10]. Our result aligns with prior research indicating that the concentrations of eucalyptol and linalool are low in fresh jujube [23]. In contrast, cis-geranylacetone exhibits high concentrations but is present only in select samples, specifically within cultivars ‘TPJDZ’, ‘LYLZ’, ‘LCYLZ’, ‘YCXZ’, ‘XZHZ’, ‘BXJZ’, and ‘ZYDZ’. Notably, cis-geranylacetone stands out as the most significant bound terpenoid, imparting rose, floral, green, and fruity aromas. Its concentration ranges from 611.50 μg/L to 637.65 μg/L among all kinds of fresh jujube, underscoring its essential role in contributing to the fruit’s aromatic profile. Overall, terpenoids significantly impact on the aroma profile of fresh jujube, demonstrating their crucial role in shaping the fruit’s olfactory characteristics.

#### 2.1.7. Phenols and Others

In this study, we detected three phenols (phenol, eugenol and 2,4-Bis (1,1-dimethylethyl) phenol) and three other compounds (3,3,5-trimethylcyclohexene, 3,4,4-trimethyl-2-cyclopenten-1-one and γ-caprolactone). Most of these compounds had a low content, except 2,4-Bis (1,1-dimethylethyl) phenol. The content of free 2,4-Bis (1,1-dimethylethyl) phenol ranged from 11.51 μg/L–68.67 μg/L, whereas the concentration of its glycoside-bound state is as high as 2096.55 μg/L (See Table 2 and Table 3).

### 2.2. Aroma Series

Despite the detection of various volatile compounds in fresh jujube samples, not all contribute significantly to the fruit’s overall aroma [34]. To assess the impact of numerous volatile compounds on the olfactory perception of fresh jujubes, we categorized the aroma compounds found in cultivars of fresh jujubes into seven groups: fruity, floral, sweet, green, fatty, earthy, and chemical. The contribution of each aroma group was quantified by summing the OAVs of the compounds within each category, provided their OAVs were greater than 1 [18]. Our analysis revealed three glycoside-bound compounds with OAVs exceeding 1 in fresh jujubes (Table 4), while twenty-four odor-active free volatiles were present at concentrations above their sensory thresholds (Table 5), markedly influencing the aroma profile of fresh jujube.

**Table 2 plants-13-01517-t002:** Concentrations (in microg/L), odor threshold, and odor descriptors of free aroma compounds in 10 jujube cultivars of various seasons.

Compounds	Category	Threshold ^a^	Descriptors ^b^	Year	JSBZ	TPJDZ	LYLZ	LCYLZ	YCXZ	TZCH	XZHZ	BXJZ	ZYDZ	CXJSXZ
Acetic acid	Acid	2.5–250	Pungent, sour, vinegar	2020	5846.46	5676.31	4391.37	3028.99	3726.11	5043.30	8016.07	8086.92	3438.95	4670.85
				2021	157.68	180.07	120.10	104.04	146.21	76.49	140.52	245.33	112.88	94.71
Butyric acid	Acid	240	Sour, cheese, butter, fruity	2020	78.66	55.36	49.58	100.67	60.79	55.26	110.81	52.19	46.79	62.81
				2021	78.73	51.42	66.68	64.39	72.14	50.29	57.90	75.52	49.00	66.70
Isovaleric acid	Acid	120–700	Sour, stinky, sweaty, cheese	2020	98.29	36.01	33.54	300.68	95.19	70.48	290.17	1382.56	82.74	293.30
				2021	27.57	tr	tr	tr	tr	tr	37.54	759.30	64.62	tr
Valeric acid	Acid	3000	Acidic	2020	54.95	64.51	59.81	69.50	58.55	58.34	64.56	62.71	76.30	55.33
				2021	98.14	75.64	90.99	73.74	87.23	73.20	103.76	139.72	109.99	58.92
Hexanoic acid	Acid	3000	Sour, fatty, cheese	2020	237.10	685.67	584.00	1194.60	331.56	189.12	395.10	533.37	379.46	287.34
				2021	997.57	1337.77	1452.96	755.07	974.28	260.64	992.05	1154.24	547.80	461.69
Octanoic acid	Acid	3000	Fatty, cheese	2020	383.89	408.55	319.31	1009.92	468.97	303.77	405.71	368.97	341.68	491.86
				2021	368.90	353.19	334.42	297.69	341.20	269.15	528.90	398.36	365.89	278.65
Nonanoic acid	Acid	3000	Cheese	2020	249.92	256.81	254.44	259.60	244.89	243.70	252.58	254.19	240.54	251.00
				2021	264.12	276.78	272.29	263.13	277.79	295.74	283.55	310.02	282.29	272.68
Decanoic acid	Acid	10,000	Rancid, sour, fatty, citrus	2020	1555.22	1216.96	1589.38	6786.03	2937.21	1864.18	2613.48	2247.90	1291.36	3029.50
				2021	927.35	813.82	876.27	1480.34	2180.98	607.21	2823.39	1801.22	962.75	1228.48
Dodecanoic acid	Acid	10,000		2020	767.04	842.12	810.03	1174.03	891.33	938.74	1020.84	934.43	591.49	1320.53
				2021	523.08	393.99	798.13	561.26	555.64	370.64	887.98	526.03	527.07	461.79
Ethyl alcohol	Alcohol			2020	7538.39	16,842.38	6261.72	1505.11	3267.59	5081.32	4032.40	9030.31	1260.89	6319.54
				2021	334.84	376.47	408.11	246.66	238.23	216.34	220.57	335.92	321.65	231.46
Pent-1-en-3-ol	Alcohol	400	Green, vegetable, tropical fruity	2020	1535.76	1604.58	211.63	158.69	86.04	79.59	63.52	137.86	240.03	66.85
				2021	689.50	821.31	543.27	591.85	895.92	182.08	402.39	809.85	651.37	220.84
Methyl-2-butan-1-ol	Alcohol	300	fruity, fusel, alcoholic	2020	40.43	99.83	50.59	35.02	33.30	34.22	42.59	65.78	34.83	55.49
				2021	32.66	35.43	34.51	32.16	0.00	31.40	34.22	36.35	16.91	16.55
Oct-1-en-3-ol	Alcohol	1	Mushroom, earthy, green	2020	1.81	5.20	4.46	2.46	3.10	1.95	3.15	4.36	2.66	2.73
				2021	2.27	3.58	2.64	1.70	3.50	0.67	4.32	3.96	2.36	2.85
2-Ethyl-1-hexanol	Alcohol	270,000	Citrus, floral, sweet	2020	2.57	2.94	3.94	2.36	2.31	2.16	2.38	1.61	1.72	4.59
				2021	1.15	0.94	0.85	1.24	0.83	1.95	1.03	1.01	1.15	1.05
Isopentanal	Aldehyde	0.2–2	Chocolate, peach, fatty	2020	37.38	123.59	50.66	22.06	15.58	16.38	30.32	312.72	18.33	51.73
				2021	10.91	12.42	12.61	10.40	12.42	9.25	13.91	150.05	10.27	13.28
(E)-But-2-en-1-al	Aldehyde		Flower	2020	9.97	7.46	7.15	7.22	9.30	7.70	7.98	7.30	7.14	8.01
				2021	tr	tr	tr	tr	tr	tr	tr	tr	tr	tr
Hexanal	Aldehyde	4.5	Grass-like, green	2020	39.30	108.47	87.59	49.12	26.00	263.13	40.54	65.81	59.09	56.99
				2021	89.43	150.63	86.46	55.63	135.92	37.43	70.67	109.72	85.54	102.04
Heptanal	Aldehyde	3	Fatty, green, herbal	2020	tr	tr	tr	tr	tr	tr	tr	tr	tr	tr
				2021	tr	11.70	10.66	tr	tr	tr	tr	tr	9.75	tr
(E)-Hex-2-en-1-al	Aldehyde	17	Green, banana, fatty, cheesy	2020	8.73	11.28	11.24	11.59	9.71	8.60	7.86	9.09	9.86	8.76
				2021	17.74	10.31	10.44	11.94	19.17	7.49	17.54	10.78	9.27	8.87
Octanal	Aldehyde	0.7	Citrus, orange, green, fatty	2020	tr	5.55	3.95	4.12	tr	4.81	7.48	5.15	2.98	tr
				2021	2.77	3.12	2.83	2.40	tr	2.70	tr	2.90	3.21	3.17
(Z)-Hept-2-en-1-al	Aldehyde	13		2020	7.27	7.99	7.55	7.30	7.39	7.35	7.45	7.48	7.42	7.40
				2021	7.30	7.29	7.21	7.13	7.29	7.08	7.42	7.32	7.25	7.47
Nonanal	Aldehyde	1	Rose, orange	2020	tr	4.33	3.54	2.60	tr	2.60	2.94	3.55	3.02	tr
				2021	4.26	5.66	4.42	3.33	2.61	4.33	2.80	4.19	6.21	4.99
(E)-2-Oct-en-1-al	Aldehyde	0.1	Cucumber, green, herbal, fatty	2020	7.62	10.16	8.98	7.56	7.70	8.16	7.82	8.45	8.22	7.94
				2021	7.19	7.61	7.40	7.20	7.25	7.24	7.38	7.56	7.60	8.05
Decanal	Aldehyde	0.1	Oil, orange, peel	2020	0.58	1.68	1.14	0.65	0.17	0.78	0.93	1.36	0.95	0.48
				2021	4.29	4.02	2.18	18.19	3.68	3.31	2.28	2.49	42.54	1.51
Non-2-en-1-al	Aldehyde	0.08–0.1	Fatty, green, cucumber	2020	2.27	2.81	2.52	2.23	2.27	2.33	2.42	2.50	2.35	2.28
				2021	2.21	2.27	2.24	3.26	2.21	2.18	2.27	2.33	4.53	2.31
(E)-Dec-2-en-1-al	Aldehyde	0.3–0.4	Fatty, earthy, coriander, green, mushroom	2020	1.35	5.77	4.63	1.79	tr	tr	1.37	2.31	3.50	1.33
				2021	tr	tr	tr	tr	1.08	1.09	1.06	1.20	1.26	1.37
Benzaldehyde	Aldehyde	350–3500	Sweet, cherry	2020	10.23	193.23	317.56	36.02	43.38	26.45	69.95	312.52	17.33	586.44
				2021	6.67	194.98	228.02	9.83	12.90	6.91	152.62	440.11	12.37	82.26
Cumaldehyde	Aldehyde	400		2020	8.36	8.58	7.90	7.09	7.46	6.37	6.19	6.43	6.28	2.86
				2021	tr	tr	2.68	tr	2.77	tr	tr	5.40	tr	tr
Pentan-3-one	Ketones	3.17–49.35	Ethereal, acetone	2020	0.00	tr	tr	107.29	tr	tr	tr	tr	tr	80.53
				2021	193.61	tr	tr	86.49	168.58	554.97	126.64	tr	132.93	120.07
Hex-4-en-3-one	Ketones		Green	2020	tr	tr	tr	tr	tr	tr	tr	tr	tr	tr
				2021	tr	tr	tr	tr	tr	tr	tr	tr	tr	tr
Pent-3-en-2-one	Ketones	15	Fruity	2020	7.23	7.46	7.16	7.07	tr	7.17	7.16	7.18	7.08	7.34
				2021	7.07	7.07	7.07	7.05	7.05	tr	3.54	7.07	7.08	7.06
Isobutyl ketone	Ketones	0.66–1.86	Green, fruity, pineapple, banana	2020	4.41	4.62	4.46	4.38	4.17	6.44	4.26	4.22	4.13	4.31
				2021	36.68	36.70	31.92	33.35	34.88	34.86	30.93	35.16	33.67	26.65
Heptan-2-one	Ketones	140	Fruity, sweet, herbal, coconut	2020	16.30	16.18	13.85	22.28	6.06	11.58	21.42	12.77	18.60	17.87
				2021	58.24	29.84	38.77	17.51	59.33	20.72	33.37	64.68	39.52	31.86
Acetoin	Ketones	800	Sweet, buttery, creamy, dairy, milky, fatty	2020	119.57	38.88	8.99	40.46	60.12	50.94	89.99	36.77	tr	74.56
				2021	tr	tr	tr	tr	tr	tr	tr	tr	tr	tr
1-Hepten-3-one	Ketones		Metallic	2020	7.38	8.56	7.78	7.37	7.55	7.51	7.69	7.61	7.46	7.56
				2021	tr	7.35	7.21	3.57	tr	7.10	7.33	7.31	7.30	7.56
2-Methyl-3-octanone	Ketones			2020	2.74	4.66	3.26	2.72	2.92	3.40	3.17	4.47	2.90	2.76
				2021	2.79	3.87	3.87	2.72	2.68	2.43	3.09	4.47	3.18	2.70
Nonan-2-one	Ketones	5–200	Sweet, green, earthy, herbal	2020	2.29	2.44	3.33	4.84	2.45	2.30	3.79	2.42	2.26	2.94
				2021	15.25	5.34	3.60	8.41	33.22	4.04	6.83	11.52	4.31	9.24
Octan-2-one	Ketones	5	Earthy, herbal	2020	1.08	2.39	1.07	1.07	tr	tr	2.30	tr	2.37	1.04
				2021	3.71	2.87	3.59	2.24	3.39	2.55	2.96	3.58	3.05	2.97
Styrene	Benzenoid	730	Sweet, balsam, floral	2020	7.00	9.22	7.16	6.70	6.65	6.58	6.92	7.55	6.85	7.70
				2021	6.24	6.01	5.91	6.01	6.41	6.47	6.00	5.98	6.05	6.85
Ethyl benzenecarboxylate	Benzenoid	60		2020	6.15	3.56	2.29	2.73	2.58	7.62	13.89	5.79	3.80	3.75
				2021	2.44	tr	tr	2.14	tr	2.16	tr	1.14	3.18	1.01
Naphthalene	Benzenoid		Pungent	2020	11.90	11.44	9.78	8.59	7.90	7.71	8.00	8.42	7.46	7.85
				2021	8.73	7.89	7.43	7.38	8.30	7.97	9.28	7.88	7.46	7.54
Analgit	Benzenoid			2020	7.15	7.23	7.00	6.74	6.92	6.54	6.52	3.29	6.51	3.19
				2021	tr	tr	tr	tr	tr	tr	tr	tr	tr	tr
Ethyl salicylate	Benzenoid	84	Sweet, mint, floral, balsam	2020	tr	tr	tr	tr	tr	tr	tr	tr	tr	tr
				2021	tr	tr	tr	tr	tr	tr	tr	tr	tr	tr
Isobutyl benzoate	Benzenoid		Sweet, fruity, musty, balsam	2020	0.92	0.93	tr	tr	tr	tr	1.93	nd	0.94	nd
				2021	1.99	tr	tr	tr	tr	tr	tr	nd	2.08	nd
Benzyl alcohol	Benzenoid	20,000	Floral, rose, balsamic	2020	36.91	145.54	203.53	nd	nd	nd	55.37	279.77	19.87	243.59
				2021	18.91	131.05	85.06	nd	nd	nd	57.21	243.02	19.53	55.12
Ethyl benzenepropanoate	Benzenoid			2020	1.90	2.42	tr	tr	tr	tr	tr	tr	tr	tr
				2021	tr	tr	tr	tr	tr	tr	tr	tr	tr	tr
β-Methylnaphthalene	Benzenoid			2020	5.91	6.11	5.89	5.81	5.95	5.86	5.98	6.12	5.88	6.00
				2021	5.66	7.72	5.65	5.68	5.70	5.73	5.74	5.79	5.64	5.65
α-Calacorene	Benzenoid			2020	6.69	9.09	7.17	8.04	5.65	8.23	8.37	8.06	tr	9.69
				2021	7.03	7.48	25.41	5.74	6.14	20.05	9.19	7.89	6.37	10.49
Phenol	Phenols		Plastic, rubber	2020	0.86	1.25	1.52	0.51	0.18	0.15	0.23	0.87	tr	0.98
				2021	0.20	6.09	0.47	0.38	1.72	0.59	1.09	1.54	0.33	0.04
Eugenol	Phenols	6	Sweet	2020	nd	3.11	3.77	nd	nd	2.99	tr	tr	2.86	tr
				2021	nd	3.02	2.92	nd	nd	tr	tr	tr	tr	tr
2,4-Bis(1,1-dimethylethyl)phenol	Phenols			2020	50.80	30.86	43.57	42.75	32.38	61.42	60.61	68.25	52.21	68.67
				2021	22.29	18.34	11.51	21.98	42.67	44.51	24.15	32.35	29.69	29.25
Methyl acetate	Esters		Sweet, fruity	2020	141.00	148.25	159.91	293.07	105.51	235.13	195.28	214.77	135.67	534.11
				2021	tr	tr	tr	tr	tr	tr	tr	tr	tr	tr
Ethyl Acetate	Esters	5000	Fruity, sweet	2020	8791.03	9775.09	3351.95	2200.36	1079.32	4631.17	5680.06	8517.00	1302.71	7985.80
				2021	54.69	65.42	98.09	97.27	49.52	44.79	62.47	173.29	66.87	46.81
Ethyl propanoate	Esters	10	Sweet, fruity, grape, pineapple	2020	tr	6.49	tr	tr	tr	3.09	2.60	3.27	tr	2.25
				2021	0.45	tr	tr	tr	tr	0.79	tr	tr	0.39	tr
Ethyl 2-methylbutanoate	Esters	0.006	Sweet, green, apple, fruity	2020	nd	18.41	9.91	nd	nd	5.22	nd	14.15	10.25	nd
				2021	nd	tr	tr	nd	nd	tr	nd	9.83	9.77	nd
Isoamyl acetate	Esters	2	Sweet, fruity, banana	2020	nd	11.62	nd	tr	nd	tr	5.09	17.15	10.41	10.73
				2021	nd	5.08	nd	10.59	nd	5.26	5.06	5.25	5.12	5.25
Ethyl hexanoate	Esters	1	Apple, peel, fruity	2020	10.94	18.21	11.42	9.90	10.69	9.88	10.64	12.50	9.29	11.22
				2021	9.54	9.89	9.84	9.52	9.56	9.60	9.31	9.59	9.50	9.43
Hexyl acetate	Esters	2	Fruity, apple, banana, green, floral	2020	1.65	1.68	1.52	1.62	tr	tr	tr	tr	tr	tr
				2021	tr	tr	tr	tr	tr	tr	tr	tr	tr	tr
Ethyl heptanoate	Esters	2.2	Fruity, pineapple	2020	tr	0.20	0.02	tr	tr	tr	tr	0.03	tr	tr
				2021	tr	tr	tr	tr	tr	tr	tr	tr	tr	tr
Ethyl lactate	Esters	14,000	Fruity, buttery	2020	tr	tr	tr	tr	tr	tr	tr	tr	tr	tr
				2021	tr	tr	tr	tr	tr	tr	tr	tr	tr	tr
Ethyl 2-hexenoate	Esters		Fruity, green, sweet	2020	tr	0.93	0.10	tr	tr	tr	tr	0.31	0.17	tr
				2021	0.78	0.13	0.09	tr	0.01	tr	0.04	0.22	0.17	0.19
Ethyl caprate	Esters		Sweet, fruity, apple, grape	2020	63.73	34.19	11.66	23.62	37.02	26.10	57.30	22.39	3.46	82.08
				2021	0.92	0.60	0.78	0.75	0.80	0.38	1.24	0.00	0.63	0.19
Ethyl dodecanoate	Esters	5900	Sweet, floral	2020	9.87	10.58	6.32	5.53	6.82	7.05	9.47	6.46	tr	12.82
				2021	tr	tr	tr	tr	tr	tr	tr	tr	tr	tr
Eucalyptol	Isoprenoids		Eucalyptus, herbal, camphor	2020	1.00	1.06	1.02	1.01	1.03	1.04	1.04	0.97	1.03	0.97
				2021	0.91	0.49	0.93	1.02	1.01	0.94	1.10	0.47	0.95	0.93
Sulcatone	Isoprenoids		Citrus, green, apple	2020	2.66	5.61	3.95	2.17	2.64	3.05	2.70	4.60	5.37	3.19
				2021	2.41	3.47	3.03	2.15	2.34	2.14	2.44	3.41	10.26	2.82
Camphor	Isoprenoids	460	Camphoreous	2020	tr	0.02	0.01	tr	tr	tr	tr	0.01	tr	tr
				2021	tr	tr	tr	tr	tr	tr	tr	tr	tr	tr
Linalool	Isoprenoids	15	Citrus, floral, sweet, rose, blueberry	2020	0.01	0.01	0.01	tr	0.01	0.01	0.01	0.01	0.01	tr
				2021	0.01	0.01	0.01	0.01	0.01	0.01	0.01	tr	0.01	tr
α-Ionene	Isoprenoids			2020	tr	1.39	0.23	0.57	tr	1.25	2.35	1.78	0.81	tr
				2021	3.57	tr	0.91	0.82	1.51	tr	2.86	0.54	1.16	0.83
Hotrienol	Isoprenoids		Sweet	2020	tr	tr	tr	tr	tr	tr	tr	tr	tr	tr
				2021	tr	tr	tr	tr	tr	tr	tr	tr	tr	tr
Levomenthol	Isoprenoids		Peppermint, minty	2020	tr	0.95	0.93	0.93	0.92	0.93	0.93	0.94	0.92	0.95
				2021	0.93	tr	tr	tr	0.90	0.93	tr	tr	0.92	0.91
α-Terpineol	Isoprenoids	330	Pine, lilac, citrus, woody, floral	2020	0.90	0.94	0.91	0.90	0.89	0.91	0.91	0.91	0.91	0.90
				2021	0.91	0.90	0.92	0.90	0.90	0.89	0.90	0.89	0.90	0.44
β-Damascenone	Isoprenoids	0.0009	Sweet, fruity, flora, honey, baked apple	2020	1.68	3.56	2.21	1.86	1.48	3.04	3.13	5.83	2.49	1.84
				2021	4.98	1.96	3.91	3.11	3.36	1.84	3.57	3.35	2.70	3.56
cis-Geranylacetone	Isoprenoids	60	Rose, floral, green, magnolia, fruity	2020	tr	tr	tr	tr	tr	tr	tr	tr	tr	tr
				2021	tr	62.05	61.88	124.87	124.43	tr	62.67	62.69	66.33	tr
3,3,5-Trimethylcyclohexene	Others			2020	0.01	0.02	0.01	0.01	0.01	0.02	0.01	0.01	0.01	0.01
				2021	tr	tr	tr	tr	tr	tr	tr	tr	tr	tr
3,4,4-Trimethyl-2-cyclopenten-1-one	Others			2020	0.01	0.02	0.01	0.01	0.01	0.02	0.01	0.01	0.01	0.01
				2021	tr	tr	tr	tr	tr	tr	tr	tr	tr	tr
γ-Caprolactone	Others		Herbal, sweet, tobacco	2020	tr	tr	tr	tr	tr	tr	tr	tr	tr	tr
				2021	tr	tr	tr	tr	tr	tr	tr	tr	tr	tr

^a^ The threshold was derived from data found on the website (https://www.vcf-online.nl). ^b^ Descriptors were sourced from http://www.thegoodscentscompany.com. ‘nd’: not detected. ‘tr’: trace amount. Odor threshold is defined as the lowest concentration of an odorant that can be detected, compared to a control with no odorant, by at least 50% of a panel of tasters (preferably more than eight individuals).

**Table 3 plants-13-01517-t003:** Concentration (in microg/L), odor threshold (in microg/L), odor descriptor of glycoside bound compounds in ten jujube cultivars.

Compounds	Category	Threshold ^a^	Describe ^b^	Year	JSBZ	TPJDZ	LYLZ	LCYLZ	YCXZ	TZCH	XZHZ	BXJZ	ZYDZ	CXJSXZ
Acetic acid	Acid	2.5–250	Pungent, sour, vinegar	2020	355.25	336.35	480.60	334.95	320.35	329.55	376.10	363.50	332.50	307.95
				2021	317.30	330.75	317.80	502.55	502.70	333.00	367.05	322.10	363.60	411.30
Formic acid	Acid	450,000	Vinegar	2020	780.95	719.25	213.85	851.75	729.05	932.15	571.10	694.30	1377.25	464.85
				2021	719.20	1157.65	768.80	1195.45	1569.40	795.55	1056.70	975.70	1006.60	854.80
Hexanoic acid	Acid	3000	Sour, fatty, cheese	2020	196.15	170.40	164.25	166.20	163.60	164.65	166.30	170.00	161.90	161.20
				2021	160.20	163.20	158.85	162.00	168.00	160.05	160.50	161.00	162.35	163.30
Octanoic acid	Acid	3000	Fatty, cheese	2020	1140.65	1144.15	1138.85	1138.15	1136.10	1140.30	1162.05	1165.35	1139.55	1131.25
				2021	1139.70	1134.95	1133.75	1140.40	1148.65	1136.55	1134.15	1134.65	1127.60	1163.55
Nonanoic acid	Acid	3000	Cheese	2020	1266.05	1316.95	1298.70	1306.30	1292.00	1299.45	1344.35	1377.15	1312.25	1264.80
				2021	1298.80	1228.05	1270.25	1239.90	1290.20	1305.90	1251.45	1276.40	1200.05	1338.15
Decanoic acid	Acid	10,000	Rancid, sour, fatty, citrus	2020	1241.65	1207.95	1199.95	1203.80	1194.60	1188.95	1212.65	1217.25	1525.00	1185.25
				2021	1200.15	1170.15	1206.15	1179.90	1199.70	1194.35	1232.50	1191.10	1133.30	1263.50
3-Methyl-butan-2-ol	Alcohol	250–300	Fruity	2020	260.10	277.20	284.60	279.50	283.15	284.50	295.20	273.15	281.80	286.25
				2021	270.50	250.15	269.50	237.45	240.05	256.70	269.05	263.45	244.00	260.35
Hexan-2-ol	Alcohol		Chemical, fruity, fatty	2020	30.40	31.30	30.90	30.85	31.55	31.40	33.05	32.80	31.50	31.35
				2021	30.55	30.30	30.35	30.55	30.15	30.10	30.70	30.25	31.10	30.55
2,7-Dimethyl-4-octanol	Alcohol			2020	1.90	1.85	1.85	1.85	1.85	1.90	1.90	1.95	1.85	1.85
				2021	1.85	1.90	1.85	2.00	2.05	1.85	1.85	1.85	2.10	1.95
4-Methyl-2-heptanol	Alcohol			2020	8.40	8.60	8.75	8.40	8.40	8.70	9.35	9.05	8.60	8.15
				2021	8.40	8.65	7.90	9.95	9.80	8.20	8.35	8.55	10.50	8.20
5-Methyl-2-heptanol	Alcohol			2020	2.30	2.35	2.40	2.30	2.35	2.40	2.45	2.45	2.35	2.35
				2021	2.30	2.30	2.25	2.50	2.50	2.30	2.30	2.35	2.70	2.30
Dodecan-5-ol	Alcohol			2020	1.80	1.80	1.80	1.80	1.80	1.85	1.90	1.90	1.85	1.80
				2021	1.80	1.80	1.75	1.95	1.95	1.80	1.80	1.80	2.10	1.90
Oct-1-en-3-ol	Alcohol	1	Mushroom, earthy, green	2020	2.20	1.60	1.25	1.60	1.30	1.35	1.50	3.20	0.95	2.00
				2021	1.60	1.55	1.20	1.85	2.50	0.95	1.60	1.75	3.00	2.85
2-Ethyl-1-hexanol	Alcohol	270,000	Citrus, floral, sweet	2020	74.90	51.30	71.70	102.00	41.55	86.45	159.50	63.30	78.70	54.25
				2021	59.90	80.35	62.35	40.80	39.35	86.45	33.60	145.50	56.90	34.15
Dodecan-1-ol	Alcohol	0.0152–0.0533	Earthy, soapy, waxy, fatty	2020	0.90	0.85	0.95	0.85	0.85	0.90	0.95	1.00	0.90	0.85
				2021	0.85	0.80	0.75	0.90	0.75	0.75	0.75	0.70	0.95	0.90
Undecan-1-ol	Alcohol		Waxy, rose, soapy, floral, citrus	2020	1.01	1.14	1.12	1.25	1.05	1.21	1.15	1.45	1.13	1.11
				2021	1.22	0.95	1.05	1.20	1.15	1.05	1.20	1.23	1.52	1.25
2,4-Bis(1,1-dimethylethyl)phenol	Phenols			2020	1748.65	1543.65	1798.60	1732.45	1722.20	2057.20	2096.55	1982.05	2256.25	1805.95
				2021	1962.50	1825.05	1714.65	2007.65	1727.50	1599.85	1847.10	1551.10	3648.40	1395.30
Hotrienol	Terpenoids		Sweet	2020	tr	tr	tr	tr	tr	tr	tr	tr	tr	tr
				2021	tr	tr	tr	tr	tr	tr	tr	tr	tr	tr
Levomenthol	Terpenoids		Peppermint, minty	2020	4.65	0.00	4.70	4.60	4.65	4.75	6.20	6.10	4.70	4.65
				2021	4.75	4.70	4.65	4.95	4.85	4.65	4.85	4.65	4.95	5.20

^a^ The threshold was derived from data found on the website (https://www.vcf-online.nl). ^b^ Descriptors were sourced from http://www.thegoodscentscompany.com. ‘nd’: not detected. ‘tr’: trace amount.

**Table 4 plants-13-01517-t004:** OAVs * of major glycoside-bound aroma compounds in 10 jujube cultivars of various seasons.

Compounds (μg/L)	Category	Threshold	Describe	Aroma Feature	Year	JSBZ	TPJDZ	LYLZ	LCYLZ	YCXZ	TZCH	XZHZ	BXJZ	ZYDZ	CXJSXZ
Acetic acid	Acid	2.5–250	Pungent, sour, vinegar	Fatty	2020	1.42	1.35	1.92	1.34	1.28	1.32	1.50	1.45	1.33	1.23
					2021	1.27	1.32	1.27	2.01	2.01	1.33	1.47	1.29	1.45	1.65
Oct-1-en-3-ol	Alcohol	1	Mushroom, earthy, green	Green, Earthy	2020	2.20	1.60	1.25	1.60	1.30	1.35	1.50	3.20		2.00
					2021	1.60	1.55	1.20	1.85	2.50		1.60	1.75	3.00	2.85
Dodecan-1-ol	Alcohol	0.0152–0.0533	Earthy, soapy, waxy, fatty	Fatty, Earthy	2020	16.89	15.95	17.82	15.95	15.95	16.89	17.82	18.76	16.89	15.95
					2021	15.95	15.01	14.07	16.89	14.07	14.07	14.07	13.13	17.82	16.89

* OAV (odor activity value): the ratio of concentration/odor threshold value.

**Table 5 plants-13-01517-t005:** OAVs of free aroma compounds in ten jujube cultivars.

Compounds (μg/L)	Category	Threshold	Describe	Aroma Feature	Year	JSBZ	TPJDZ	LYLZ	LCYLZ	YCXZ	TZCH	XZHZ	BXJZ	ZYDZ	CXJSXZ
Acetic acid	Acid	2.5–250	Pungent, sour, vinegar	Fatty	2020	23.39	22.71	17.57	12.12	14.90	20.17	32.06	32.35	13.76	18.68
					2021	15.77	18.01	12.01	10.40	14.62	7.65	14.05	24.53	11.29	9.47
Isovaleric acid	Acid	120–700	Sour, stinky, sweaty, cheese	Fatty	2020	0.00	0.00	0.00	0.00	0.00	0.00	0.00	1.98	0.00	0.00
					2021	0.00	0.00	0.00	0.00	0.00	0.00	0.00	1.08	0.00	0.00
Pent-1-en-3-ol	Alcohol	400.00	Green, vegetable, tropical fruity	Fruity, green	2020	3.84	4.01	0.00	0.00	0.00	0.00	0.00	0.00	0.00	0.00
					2021	1.72	2.05	1.36	1.48	2.24	0.00	1.01	2.02	1.63	0.00
Oct-1-en-3-ol	Alcohol	1.00	Mushroom, earthy, green	Green, earthy	2020	1.81	5.20	4.46	2.46	3.10	1.95	3.15	4.36	2.66	2.73
					2021	2.27	3.58	2.64	1.70	3.50	0.00	4.32	3.96	2.36	2.85
Isopentanal	Aldehyde	0.2–2	Chocolate, peach, fatty	Fruity, fatty	2020	18.69	61.80	25.33	11.03	7.79	8.19	15.16	156.36	9.17	25.87
					2021	5.46	6.21	6.31	5.20	6.21	4.63	6.96	75.03	5.14	6.64
Pentanal	Aldehyde	12.00	Fruity, nutty, berry	Fruity	2020	5.06	15.61	11.65	7.27	3.26	8.08	5.70	13.14	6.08	5.31
					2021	0.00	25.28	18.80	4.94	0.00	0.00	0.00	22.96	10.77	10.54
Hexanal	Aldehyde	4.50	Grass-like, green	Green	2020	8.73	24.10	19.46	10.92	5.78	58.47	9.01	14.62	13.13	12.66
					2021	19.87	33.47	19.21	12.36	30.20	8.32	15.70	24.38	19.01	22.68
Heptanal	Aldehyde	3.00	Fatty, green, herbal	Green, fatty	2020	0.00	0.00	0.00	0.00	0.00	0.00	0.00	0.00	0.00	0.00
					2021	0.00	3.90	3.55	0.00	0.00	0.00	0.00	0.00	3.25	0.00
(E)-Hex-2-en-1-al	Aldehyde	17.00	Green, banana, fatty, cheesy	Fruity, green, fatty	2020	0.00	0.00	0.00	0.00	0.00	0.00	0.00	0.00	0.00	0.00
					2021	1.04	0.00	0.00	0.00	1.13	0.00	1.03	0.00	0.00	0.00
Octanal	Aldehyde	0.70	Citrus, orange, green, fatty	Fruity, green, fatty	2020	0.00	7.93	5.64	5.89	0.00	6.87	10.69	7.36	4.26	0.00
					2021	3.96	4.46	4.04	3.43	0.00	3.86	0.00	4.14	4.59	4.53
Nonanal	Aldehyde	1.00	Rose, orange	Fruity, floral	2020	0.00	4.33	3.54	2.60	0.00	2.60	2.94	3.55	3.02	0.00
					2021	4.26	5.66	4.42	3.33	2.61	4.33	2.80	4.19	6.21	4.99
(E)-2-Oct-en-1-al	Aldehyde	0.10	Cucumber, green, herbal, fatty	Green, fatty	2020	76.20	101.60	89.80	75.60	77.00	81.60	78.20	84.50	82.20	79.40
					2021	71.90	76.10	74.00	72.00	72.50	72.40	73.80	75.60	76.00	80.50
Decanal	Aldehyde	0.10	Oil, orange, peel	Fruity, fatty	2020	5.80	16.80	11.40	6.50	1.70	7.80	9.30	13.60	9.50	4.80
					2021	42.90	40.20	21.80	181.90	36.80	33.10	22.80	24.90	425.40	15.10
Non-2-en-1-al	Aldehyde	0.08–0.1	Fatty, green, cucumber	Green, fatty	2020	22.70	28.10	25.20	22.30	22.70	23.30	24.20	25.00	23.50	22.80
					2021	22.10	22.70	22.40	32.60	22.10	21.80	22.70	23.30	45.30	23.10
(E)-Dec-2-en-1-al	Aldehyde	0.3–0.4	Fatty, earthy, coriander, green, mushroom	Green, fatty, earthy	2020	3.38	14.43	11.58	4.48	0.00	0.00	3.43	5.78	8.75	3.33
					2021	0.00	0.00	0.00	0.00	2.70	2.73	2.65	3.00	3.15	3.43
Pentan-3-one	Ketones	3.17–49.35	Ethereal, acetone	Chemical	2020	0.00	0.00	0.00	2.17	0.00	0.00	0.00	0.00	0.00	1.63
					2021	3.92	0.00	0.00	1.75	3.42	11.25	2.57	0.00	2.69	2.43
Isobutyl ketone	Ketones	0.66–1.86	Green, fruity, pineapple, banana	Fruity, green	2020	2.37	2.48	2.40	2.35	2.24	3.46	2.29	2.27	2.22	2.32
					2021	19.72	19.73	17.16	17.93	18.75	18.74	16.63	18.90	18.10	14.33
Methyl benzoate	Benzene	0.52	Floral	Floral	2020	0.00	0.00	0.00	0.00	0.00	0.00	4.21	0.00	0.00	0.00
					2021	0.00	0.00	0.00	0.00	0.00	0.00	0.00	0.00	0.00	0.00
Ethyl Acetate	Esters	5000.00	Fruity, sweet	Fruity, sweet	2020	1.76	1.96	0.00	0.00	0.00	0.00	1.14	1.70	0.00	1.60
					2021	0.00	0.00	0.00	0.00	0.00	0.00	0.00	0.00	0.00	0.00
Ethyl 2-methylbutanoate	Esters	0.01	Sweet, green, apple, fruity	Fruity, sweet, green	2020	0.00	3068.33	1651.67	0.00	0.00	870.00	0.00	2358.33	1708.33	0.00
					2021	0.00	0.00	0.00	0.00	0.00	0.00	0.00	1638.33	1628.33	0.00
Isoamyl acetate	Esters	2.00	Sweet, fruity, banana	Fruity, sweet	2020	0.00	5.81	0.00	0.00	0.00	0.00	2.55	8.58	5.21	5.37
					2021	0.00	2.54	0.00	5.30	0.00	2.63	2.53	2.63	2.56	2.63
Ethyl hexanoate	Esters	1.00	Apple, peel, fruity	Fruity	2020	10.94	18.21	11.42	9.90	10.69	9.88	10.64	12.50	9.29	11.22
					2021	9.54	9.89	9.84	9.52	9.56	9.60	9.31	9.59	9.50	9.43
β-Damascenone	Terpenoids	0.00	Sweet, fruity, flora, honey, baked apple	Fruity, floral, sweet	2020	1866.67	3955.56	2455.56	2066.67	1644.44	3377.78	3477.78	6477.78	2766.67	2044.44
					2021	5533.33	2177.78	4344.44	3455.56	3733.33	2044.44	3966.67	3722.22	3000.00	3955.56
cis-Geranylacetone	Terpenoids	60.00	Rose, floral, green, magnolia, fruity	Fruity, floral, green	2020	0.00	0.00	0.00	0.00	0.00	0.00	0.00	0.00	0.00	0.00
					2021	0.00	1.03	1.03	2.08	2.07	0.00	1.04	1.04	1.11	0.00

In our analysis depicted in Figure 1, the fruity aroma emerged as the predominant aromatic feature across the fresh jujube cultivars, primarily attributed to the presence of free aldehydes, esters, and terpenoids. This fruity characteristic encompasses six aldehydes (isopentanal, pentanal, (E)-hex-2-en-1-al, octanal, nonanal, and decanal), four esters (ethyl acetate, ethyl 2-methylbutanoate, isoamyl acetate, and ethyl hexanoate), two terpenoids (β-damascenone and cis-geranylacetone), one alcohol (pent-1-en-3-ol), and one ketone (isobutyl ketone). The odor activity value (OAV) for the fruity aroma in fresh jujube ranged from 1670.33 (YCXZ) to 7162.82 (TPJDZ) in 2020. Notably, pentanal exhibited a higher OAV in the ‘TPJDZ’ and ‘LYLZ’ cultivars in both 2020 and 2021, characterized by its fruity, nutty, and berry aroma notes.

Floral and sweet aromas were also identified as key aromatic aspects of fresh jujubes. The floral aroma is composed of two terpenoids (β-damascenone and cis-geranylacetone), one aldehyde (nonanal), and one benzene compound (methyl benzoate). In 2020, the ‘BXJZ’ samples had the highest floral OAV at 6481.33, with ‘YCXZ’ exhibiting the lowest at 1644.44. In the following year, the highest floral OAV was recorded at 5537.59 (JSBZ), and the lowest at 2048.77 (TZCH). The sweet aroma profile includes three esters (ethyl acetate, ethyl 2-methylbutanoate, and isoamyl acetate) and one terpenoid (β-damascenone), with OAVs ranging from 1644.44 (YCXZ) to 8846.39 (BXJZ) in 2020, and from 2180.32 (TPJDZ) to 5533.33 (JSBZ) in 2021. Ethyl 2-methylbutanoate, noted for its fruity, sweet, and green flavor notes, has a remarkably low threshold of 0.006 μg/L. ‘BXJZ’ was distinguished by its high levels of fruity, floral, and sweet aroma characteristics in both 2020 and 2021, attributed to the significant presence of ethyl 2-methylbutanoate and β-damascenone. It is essential to highlight that β-damascenone, with its wide array of sweet, fruity, floral, honey, baked, and apple flavor notes and a high OAV (ranging from 1866.67 to 5533.33 in our results), plays a critical role as one of the most influential aroma compounds. It significantly contributes to the floral, fruity, and sweet aroma profile of all the fresh jujube cultivars studied.

These fresh jujube cultivars also exhibited the green and fatty flavors. The green aroma consisted of seven aldehydes (hexanal, heptanal, (E)-hex-2-en-1-al, octanal, (E)-2-oct-en-1-al, non-2-en-1-al, and (E)-dec-2-en-1-al), two alcohols (pent-1-en-3-ol and oct-1-en-3-ol), one ketone (isobutyl ketone), one ester (ethyl 2-methylbutanoate), and one terpenoid (cis-geranylacetone). The cultivar ‘TPJDZ’ showed a notably high green aroma OAV (187.85 in 2020 and 167.03 in 2021, respectively), attributed to its significant OAVs of hexanal, (E)-2-oct-en-1-al, and non-2-en-1-al. The cultivars ‘BXJZ’ and ‘ZYDZ’ exhibited higher values in fruity, sweet, and green aromas, due to the flavor contribution from ethyl 2-methylbutanoate. The fatty flavor profile included eight aldehydes (isopentanal, heptanal, (E)-hex-2-en-1-al, octanal, (E)-2-oct-en-1-al, decanal, non-2-en-1-al, and (E)-dec-2-en-1-al) and two acids (acetic acid and isovaleric acid). The cultivars ‘YCXZ’ and ‘CXJSXZ’ had the lowest OAV for green aroma in 2020 and 2021. It is important to note that (E)-2-oct-en-1-al and non-2-en-1-al, with OAVs higher than 20 in fresh jujube, are likely to significantly contribute to the green and fatty characteristics, offering cucumber, green, herbal, and fatty aromatic notes. These results are in line with previous studies [18].

The earthy and chemical attributes contributed negatively to the overall aroma profile of fresh jujube, in contrast to other aromatic characteristics. The earthy feature was characterized by one alcohol (oct-1-en-3-ol) and one aldehyde ((E)-dec-2-en-1-al). The cultivar ‘TZCH’ exhibited a low OAV for the earthy aroma, registering 1.95 in 2020 and 2.73 in 2021. The chemical aroma was defined by one ketone (pentan-3-one), with not all cultivars displaying this chemical attribute.

The results of our sensory perception prediction indicate that fruity, floral, and sweet aromas constitute the predominant aromatic profiles of the presently studied fresh jujubes, accompanied by notable green and fatty characteristics. This observation is in general agreement with the conclusions of earlier studies, which have highlighted green, fruity, floral, sweet, and fatty features as the principal sensory attributes affecting fresh jujube’s sensory evaluation [13]. Thus, our predictive analysis corroborates these prior findings.

The potential aroma contribution from the glycoside-bound volatiles in these ten cultivars of fresh jujube was not significant. The flavor feature of glycoside-bound compounds was roughly similar in these cultivars. Fatty, green and earthy were the main bound aroma features; however, differences were found in the content of aroma compounds of each cultivars. Nevertheless, only three bound compounds had their concentration above their threshold, including acetic acid, oct-1-en-3-ol, and dodecan-1-ol. Fatty attributes were mainly due to acetic acid and dodecan-1-ol, with their OAV above 1. ‘LYLZ’ had the highest OAV of bound dodecan-1-ol up to 17.82 and 16.89 in 2020 and 2021, respectively. The green feature included 1 alcohol (oct-1-en-3-ol), whereas earthy attributes were mainly due to dodecan-1-ol and oct-1-en-3-ol. Few bound compounds were detected in jujube, and only three compounds make a potential contribution to jujubes’ overall aroma when their concentration exceeds their threshold. This shows that the potential contribution of glycoside bound aroma in jujube was very limited.

The investigation into the potential aromatic contributions of glycoside-bound volatiles in ten fresh jujube cultivars revealed a minimal impact. The flavor profiles of these glycoside-bound compounds exhibited a notable uniformity across the cultivars, characterized predominantly by fatty, green, and earthy notes (Table 4). Despite this overarching similarity, variances in the concentrations of specific aroma compounds were observed among the different cultivars. Notably, only three bound compounds (acetic acid, oct-1-en-3-ol, and dodecan-1-ol) exhibited concentrations surpassing their respective threshold values, indicating a potential contribution to the overall aroma profile. The fatty attributes were predominantly attributed to acetic acid and dodecan-1-ol, both of which demonstrated OAV exceeding 1. Specifically, the cultivar ‘LYLZ’ exhibited the most pronounced OAV for bound dodecan-1-ol, with values reaching 17.82 and 16.89 in 2020 and 2021, respectively. The green aromatic feature was associated with the alcohol oct-1-en-3-ol, while the earthy attributes were primarily due to dodecan-1-ol and oct-1-en-3-ol. In jujube, a limited number of glycoside-bound compounds were identified, with merely three of these compounds exhibiting the potential to significantly contribute to the overall aroma profile. Consequently, this constrains the role of glycosidically bound compounds in enhancing the aromatic profile of fresh jujube.

### 2.3. Correlation between Volatiles and Sensory Attributes in Jujube Using PLSR

Partial least squares regression analysis (PLSR) was further utilized in the present study to elucidate the correlation between the volatile compounds (OAV > 1 for volatiles or OAV > 0.1 for esters) and sensory features in these jujube samples (Figure 2). The correlation loading in the PLSR analysis included 21 volatiles determined by GC-MS and 5 odor features (jujube aroma and sour aroma). It was found that the jujube note was positively correlated with octanal, (E)-2-oct-en-1-al, nonanal, β-damascenone, and pentanal (Figure 2). These compounds are known for their pivotal roles in shaping the odor profile of various fruits, contributing to a spectrum of olfactory sensations. Octanal and nonanal, with their fruity, green, floral notes, are crucial in imparting a fresh and fruit aroma to the jujube fruit, reminiscent of ripe citrus fruits. (E)-2-Oct-en-1-al has fruity and green notes, adding depth and complexity to the fruit’s overall aroma. β-damascenone is renowned for its powerful fruity and floral scent, which at low concentrations, can introduce a subtle richness to the jujube’s aroma, evoking the sweet and fruity smell of jujube fruits. Lastly, pentanal contributes with its slightly fruity, nutty, and berry notes, rounding off the aroma profile with a hint of berry fragrance. Together, these compounds synergize to define the distinctive fruity aroma of jujube fruits, influencing the sensory perception and appeal of the fruit. Their presence and relative concentrations are instrumental in distinguishing the aroma profile of jujubes from other fruits, highlighting the intricate relationship between volatile organic compounds and the sensory characteristics of food. Through partial least squares regression (PLSR) analysis, our study suggests that these carbonyl compounds are the primary contributors to the aroma in jujubes. This finding explains why the fragrance of fresh jujube fruits is subtler compared to other fruits.

## 3. Materials and Methods

### 3.1. Materials

In this investigation, ten fresh jujube cultivars were employed from the Research Institute of Pomology, affiliated with the Shanxi Academy of Agricultural Sciences, China. The cultivars included ‘Jishanbanzao’ (JSBZ), ‘Tupujidan’ (TPJD), ‘Linyilizao’ (LYLZ), ‘Liaochengyuanling’ (LCYL), ‘Yunchengxiangzao’ (YCXZ), ‘Tengzhouchanghong’ (TZCH), ‘Xinzhenghuizao’ (XZHZ), ‘Binxianjinzao’ (BXJZ), ‘Zhongyangmuzao’ (ZYMZ), and ‘Jinsixiaozao’ (JSXZ). The fruit was collected at its semi-red stage and immediately transported to the laboratory under a cold chain protocol at 4 °C. The experiments were conducted over two consecutive years, 2020 and 2021.

### 3.2. Reagents

Distilled water was provided from the Milli-Q pure water system (Midbury, Bedford, MA, USA). Dichloromethane, ethanol, and methanol were purchased from Honeywell (Mosley, NJ, USA). Malic acid was purchased from Sigma Aldrich (St. Louis, MO, USA). Glucose, sodium hydroxide, sodium chloride, citric acid, and sodium dihydrogen phosphate came from Beijing Chemical Plant (Beijing, China). The chemical standards were purchased from Sigma-Aldrich (St. Louis, MO, USA) with purity above 98%.

### 3.3. Sample Pre-Treatment

The collected samples were peeled and de-nucleated, and then frozen in liquid nitrogen. At low temperature, the jujube fruits were quickly crushed into powder by grinder machine. The crushed samples were stored at −80 °C for subsequent analysis. An amount of 60 g of sample powder was weighed and dispensed in a 50 mL centrifuge tube. the sample was then added with distilled water for immersion at 4 °C for 24 h and centrifuged at 8000 r/min for 15 min at low temperature (4 °C). The supernatant was transferred into the PET bottle, and placed in the refrigerator at 4 °C [34].

### 3.4. Free Volatile Extraction 

The headspace solid phase microextraction (HS-SPME) method for the extraction of aroma components is based on that of Wang et al. (2015) and Liu et al. (2018) with minor modifications [18,35]. Sample supernatant (5 mL) was transferred to a 15 mL vial containing a magnetic stir bar. Meanwhile 1.00 g of NaCl and 10 μL of internal standard (4-methyl-2-pentanol, 1.0388 g/L) were added for accuracy, and then tightened with a vial cap made of PTFE. The vial was then placed on a magnetic stirring heating table while the temperature was held at 40 °C for 30 min. The activated SPME extraction head was then inserted into the vial, which was 1 cm above from the liquid level. In the condition of heating and stirring at 40 °C, adsorption for 30 min was needed to make the aroma components balanced among the liquid, the headspace, and the SPME extraction head. The extraction head was removed and immediately inserted into the GC inlet at 250 °C for 8 min.

### 3.5. Bound Volatile Extraction

Extraction of bound aroma compounds in jujubes followed previously reported methods [35,36]. Briefly, extraction was performed by adsorption of glycosides on Cleanert PEP-SPE resins (200 mg/mL, Bonna-Agela Technologies, Tianjin, China). First, the resin column was pretreated with 10 mL methanol and 10 mL water, respectively. Subsequently, 5 mL of sample supernatant was used to add to the resin column and washed with 5 mL of water to remove sugars and organic acids. Next, it was eluted with 10 mL dichloromethane under a flow rate of 2 mL/min to remove free aroma compounds. The glycosides in the resin column were eluted with 20 mL methanol. Methanol was concentrated to dryness using a rotary evaporator at 45 °C, and then the dryness was re-dissolved in 5 mL of 2 M citrate–phosphate buffer solution (pH = 5.0). After that, 100 μL of AR2000 (Rapidase, DSM Food Specialties, Seclin, France) enzyme solution (100 mg/mL in 2 M citrate–phosphate buffer, pH 5.0) was added. It was placed in a 40 °C incubator for 16 h to perform enzymatic hydrolysis, thus liberating the free aroma. The SPME method was the same as that used for the free volatiles’ analysis.

### 3.6. GC-MS Analysis

The method of gas chromatography–mass spectrometry for the analysis of aroma substances were as per our previous reports [18,37]. The gas chromatograph used in the experiment was an Agilent 7890 GC, while the mass spectrum was Agilent 5975B MS (Agilent, Santa Clara, CA, USA). The capillary column used was HP-INNOWAX (60 m × 0.25 mm × 0.25 μm, J & W scientific, Folsom, CA, USA). High-purity helium was used as the carrier gas, whose flow rate was 1 mL/min. Solid phase micro-extraction manual injection used the splitless mode, inserting into the gas chromatographic inlet which was 250 °C for 8 min. The procedure was maintained at 50 °C for 1 min and then ramped to 220 °C at 3℃/min for 5 min. The mass spectrometry interface temperature was 280 °C. The ion source temperature was 230 °C. The ionization mode was EI, the ionization energy was 70 ev, and the mass scan range was 20–350 u.

Mass spectrometry full ion scans were used. The retention index of the compound was calculated by analyzing the C6–C24 n-alkane retention index under the same chromatographic conditions. The results of the mass spectrometry were searched and matched with the NIST11 standard library, and the obtained compound standard retention index was used to identify others. For the compounds without the standard, the semi-qualitative analysis of the retention index of the compound and the NIST11 standard library alignment in the similar chromatographic conditions reported in the literature were used. For the compounds for which standards were available, qualitative analysis was performed according to their retention indices and mass spectrum with the NIST11 standard library alignment results [34].

### 3.7. Quantitative and OAVs Calculation

In this study, the quantification of volatile compounds was performed using our previously published method with minor modifications. Based on the current physicochemical indices of jujube fruit, a current matrix prepared from 170 g/L sugar and 3.5 g/L citric acid was adopted. Its pH was adjusted to 3.5 using a 5 M NaOH solution. All standard was dissolved in HPLC-grade ethanol and used as a reserve standard solution. It was mixed with a synthetic current matrix to obtain a working standard solution. The working standard was then diluted working standard solution continuously to 18 continuous levels. Then, the working standard solution was extracted and analyzed by GC-MS using the same method as jujube liquid [18].

The standard curve was integrated by the peak area ratio of external standard to internal standard and the concentration of external standard. The regression coefficients of the quantitative standard curve were all above 95%. The aroma substance in the jujube sample was quantitatively analyzed by standard curve with existing standard compound, and aroma substances where no standards were available were quantified by the principle that the chemical structure and the number of carbon atoms were similar. The concentrations of bound volatiles were quantified by calibrating against their standard curves [35].

### 3.8. Sensory Evaluation

The sensory attributes of jujube samples were assessed through a structured evaluation conducted by a professional panel, consisting of 12 members (6 males and 6 females) within the age range from 21 to 35 years. Each panelist was tasked to characterize the aroma of the jujubes, rating their intensity on a 9-point scale, from 1 to 9 points. Ultimately, the aroma profile of the jujubes was predominantly identified as a blend of sweet and sour flavors. To ensure unbiased evaluation, the jujube samples were anonymized using a random coding system and presented in a random sequence. Panelists employed the 9-point scale to judge the flavor and acidity levels of the fresh jujube samples. A mandatory 30 s interval was observed between the tastings of each sample, with each sample being evaluated twice to ensure the reliability of the process [38].

### 3.9. Statistical Analysis Method

Mass spectrometry full ion scan (Scan) spectra were quantified using MSD Chemstation Data Analysis software F.01.01 (Agilent, Santa Clara, CA, USA). The experimental data were expressed in the form of ± standard deviation. Univariate variables analysis (ANOVA) was performed using the SPSS 22.0 (for Windows SPSS Inc., Chicago, IL, USA) for Duncan’s multiple range test with a minimum significant level of *p* < 0.05. Principal component analysis and cluster analysis were performed by XLSTAT software 2019.2.2 (Addinsoft, Paris, France).

## 4. Conclusions

This research has carefully elucidated the intricate aroma profile of fresh jujube fruits. Through the adept application of gas chromatography–mass spectrometry (GC-MS), this study has identified a broad spectrum of volatile free and glycosidically bound aroma compounds, with esters, aldehydes, and ketones standing out as the predominant classes. In our study, we conducted a comprehensive investigation into the glycosidically bound compounds present in fresh jujube fruits for the first time, identifying a total of 19 such compounds. We identified eight ketone compounds in fresh jujube cultivars for the first time, including hex-4-en-3-one, pent-3-en-2-one, isobutyl ketone, heptan-2-one, acetoin, 1-hepten-3-one, 2-methyl-3-octanone, and octan-2-one. Additionally, we detected 10 types of benzenoid aromatic compounds; these compounds are styrene, methyl benzoate, ethyl benzenecarboxylate, naphthalene, analgit, isobutyl benzoate, benzyl alcohol, ethyl benzenepropanoate, β-methylnaphthalene, and α-calacorene.

Odor activity value (OAV) analysis has been pivotal in delineating the primary aroma attributes of jujube fruits, with β-damascenone emerging as a critical contributor to the fruity aroma, while (E)-2-oct-en-1-al and nonanal have been identified as key influencers of the fatty aroma. Furthermore, the integration of sensory evaluation and partial least squares regression (PLSR) analysis has sharpened our understanding of the jujube’s characteristic aroma, highlighting octanal, (E)-2-oct-en-1-al, nonanal, β-damascenone, and pentanal as significant contributors, and revealing the notable impact of isoamyl acetate on the fatty acid taste.

This investigation sheds light on the complexity and variability of the jujube fruit’s aroma composition. The findings from this study not only enhance our comprehension of the jujube’s aroma profile but also offer valuable insights for the agricultural and food industry, paving the way for the development of improved cultivars and cultivation techniques that can optimize aroma profiles for consumer satisfaction.

## Figures and Tables

**Figure 1 plants-13-01517-f001:**
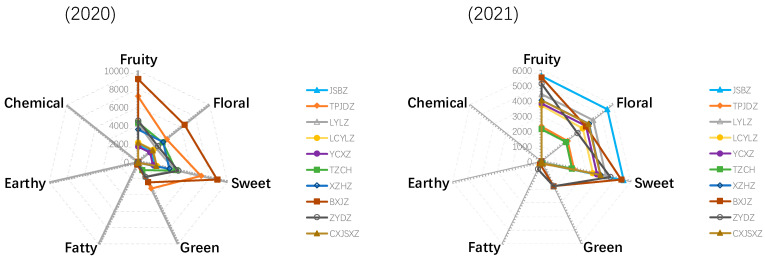
Aroma radar of ten jujubes in 2020 and 2021 year with total OAVs (∑OAV > 1) of aroma series in free volatile compounds (OAV > 1).

**Figure 2 plants-13-01517-f002:**
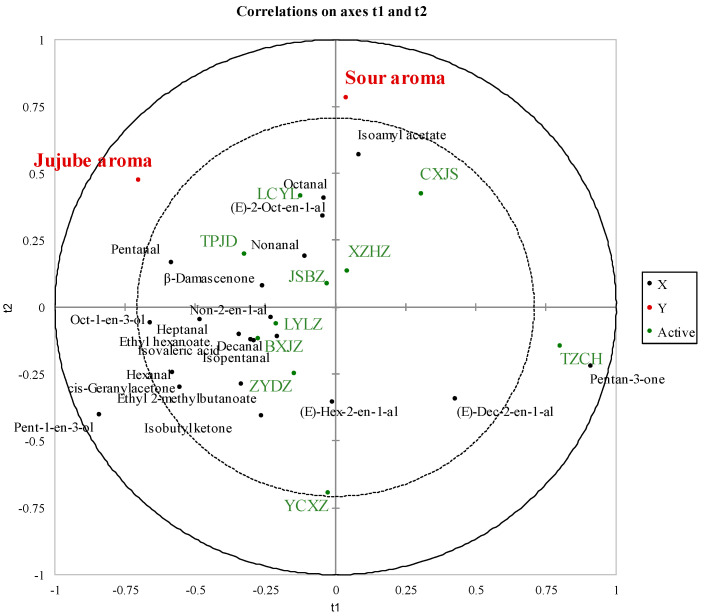
Partial least squares regression (PLSR) analysis using volatile compounds (OAV > 1 or esters OAV > 0.1) and aroma descriptors in jujube in 2021.

**Table 1 plants-13-01517-t001:** Qualitative and quantitative information of compounds.

Compounds (μg/L)	Category	Standard	Qualitative ^a^	Curve	RI ^b^	Quantify ^c^
Acetic acid	Acid	Butyric Acid	B	y = 21,865x + 42.766	1433	43
Butyric acid	Acid	Butyric Acid	A	y = 21,865x + 42.766	1620	60
Isovaleric acid	Acid	Butyric Acid	B	y = 21,865x + 42.766	1660	60
Formic acid	Acid	Butyric Acid	B	y = 21,865x + 42.766	1494	46
Valeric acid	Acid	Butyric Acid	B	y = 21,865x + 42.766	1711	60
Hexanoic acid	Acid	Hexanoic acid	A	y = 8519.9x + 30.542	1829	60
Octanoic acid	Acid	Octanoic Acid	A	y = 35,255x + 219.26	2035	60
Nonanoic acid	Acid	Octanoic Acid	B	y = 35,255x + 219.26	2138	60
Decanoic acid	Acid	Octanoic Acid	B	y = 35,255x + 219.26	2265	60
Dodecanoic acid	Acid	Octanoic Acid	B	y = 35,255x + 219.26	2503	73
3-Methyl-butan-2-ol	Alcohol	Isopentanol	B	y = 19,779x + 31.211	1100	45
Hexan-2-ol	Alcohol	Hexan-1-ol	B	y = 1546.3x − 5.0952	1219	45
2,7-Dimethyl-4-octanol	Alcohol	Octan-3-ol	B	y = 238.79x + 0.2758	1336	69
4-Methyl-2-heptanol	Alcohol	Octan-3-ol	B	y = 238.79x + 0.2758	1349	45
5-Methyl-2-heptanol	Alcohol	Octan-3-ol	B	y = 238.79x + 0.2758	1353	45
Dodecan-5-ol	Alcohol	Octan-3-ol	B	y = 238.79x + 0.2758	1390	69
Ethyl alcohol	Alcohol	Isopentanol	B	y = 19,779x + 31.211	930	31
Pent-1-en-3-ol	Alcohol	Isopentanol	B	y = 19,779x + 31.211	1164	57
Methyl-2-butan-1-ol	Alcohol	Isopentanol	B	y = 19,779x + 31.211	1208	70
Oct-1-en-3-ol	Alcohol	(E)-Hex-3-en-1-ol	B	y = 728.12x + 0.0508	1428	57
2-Ethyl-1-hexanol	Alcohol	Octan-3-ol	B	y = 238.79x + 0.2758	1466	57
Dodecan-1-ol	Alcohol	Octan-1-ol	B	y = 70.259x + 0.1283	1953	43
Undecan-1-ol	Alcohol	Octan-1-ol	B	y = 70.259x + 0.1283	1889	69
Isopentanal	Aldehyde	Hexanal	B	y = 7081x + 8.15	906	58
(E)-But-2-en-1-al	Aldehyde	(E)-Hex-2-en-1-al	B	y = 117.26x +7.04	1062	70
Hexanal	Aldehyde	Hexanal	A	y = 7081x + 8.15	1098	56
Heptanal	Aldehyde	Hexanal	B	y = 7081x + 8.15	1199	57
(E)-Hex-2-en-1-al	Aldehyde	(E)-Hex-2-en-1-al	A	y = 117.26x +7.04	1228	41
Octanal	Aldehyde	Nonanal	B	y = 648.61x + 1.9932	1287	43
(Z)-Hept-2-en-1-al	Aldehyde	(E)-Hex-2-en-1-al	B	y = 117.26x + 7.04	1321	41
Nonanal	Aldehyde	Nonanal	A	y = 648.61x + 1.9932	1382	57
(E)-2-Oct-en-1-al	Aldehyde	(E)-Hex-2-en-1-al	B	y = 117.26x + 7.04	1416	41
Decanal	Aldehyde	Decanal	A	y = 2272.1x − 0.2855	1491	43
Non-2-en-1-al	Aldehyde	Nonanal	B	y = 648.61x + 1.9932	1516	43
(E)-Dec-2-en-1-al	Aldehyde	Nonanal	B	y = 648.61x + 1.9932	1646	43
Benzaldehyde	Aldehyde	Benzaldehyde	A	y = 1782x + 1.4342	1512	106
Cumaldehyde	Aldehyde	Styrene	B	y = 1228x + 5.2634	1781	133
Pentan-3-one	Ketones	Hexanal	B	y = 7081x + 8.15	996	57
Hex-4-en-3-one	Ketones	(E)-Hex-2-en-1-al	B	y = 117.26x +7.04	1197	69
Pent-3-en-2-one	Ketones	(E)-Hex-2-en-1-al	B	y = 117.26x +7.04	1139	69
Isobutyl ketone	Ketones	Nonanal	B	y = 648.61x + 1.9932	1168	57
Heptan-2-one	Ketones	Hexanal	B	y = 7081x + 8.15	1195	43
Acetoin	Ketones	Hexanal	B	y = 7081x + 8.15	1285	45
1-Hepten-3-one	Ketones	(E)-Hex-2-en-1-al	B	y = 117.26x + 7.04	1298	55
2-Methyl-3-octanone	Ketones	Nonanal	B	y = 648.61x + 1.9932	1318	43
Nonan-2-one	Ketones	Nonanal	B	y = 648.61x + 1.9932	1379	43
Octan-2-one	Ketones	Nonanal	B	y = 648.61x + 1.9932	1283	43
Styrene	Benzenoid	Styrene	A	y = 1228x + 5.2634	1262	104
Ethyl benzenecarboxylate	Benzenoid	Ethyl salicylate	B	y = 537.33x + 1.7861	1666	105
Naphthalene	Benzenoid	Styrene	B	y = 1228x + 5.2634	1741	128
Analgit	Benzenoid	Analgit	A	y = 378.52x + 6.2031	1755	120
Ethyl salicylate	Benzenoid	Ethyl salicylate	A	y = 537.33x + 1.7861	1793	120
Isobutyl benzoate	Benzenoid	Ethyl salicylate	B	y = 537.33x + 1.7861	1835	105
Benzyl alcohol	Benzenoid	Benzyl alcohol	A	y = 36338x + 36.142	1844	108
Ethyl benzenepropanoate	Benzenoid	Ethyl salicylate	B	y = 537.33x + 1.7861	1862	104
β-Methylnaphthalene	Benzenoid	Styrene	B	y = 1228x + 5.2634	1872	142
α-Calacorene	Benzenoid	Styrene	B	y = 1228x + 5.2634	1887	157
Phenol	phenols	Phenol	A	y = 5832.8x − 1.562	2011	94
Eugenol	phenols	p-Ethylguaiacol	B	y = 729.32x + 2.8065	2156	164
2,4-Bis(1,1-dimethylethyl)phenol	phenols	Phenol	B	y = 5832.8x − 1.562	2315	191
Methyl acetate	Esters	Ethyl Acetate	B	y = 8635.7x + 18.973	804	43
Ethyl Acetate	Esters	Ethyl Acetate	A	y = 8635.7x + 18.973	878	43
Ethyl propanoate	Esters	Propyl acetate	B	y = 1187.9x − 0.515	964	57
Ethyl 2-methylbutanoate	Esters	Isoamyl acetate	B	y = 948.91x + 9.6091	1069	102
Isoamyl acetate	Esters	Isoamyl acetate	A	y = 948.91x + 9.6091	1132	43
Ethyl hexanoate	Esters	Ethyl hexanoate	A	y = 628.86x + 9.0529	1236	88
Hexyl acetate	Esters	Hexyl acetate	A	y = 631.81x + 1.396	1269	43
Ethyl heptanoate	Esters	Ethyl heptanoate	A	y = 511.01x − 0.0218	1325	88
Ethyl lactate	Esters	Ethyl butanoate	B	y = 1624.7x + 6.7119	1334	45
Ethyl 2-hexenoate	Esters	Ethyl 2-hexenoate	A	y = 521.8x − 0.1164	1340	55
Ethyl caprate	Esters	Ethyl caprate	A	y = 3321.8x − 0.1688	1635	88
Ethyl dodecanoate	Esters	Ethyl caprylate	B	y = 1049.9x + 4.7153	1819	88
Eucalyptol	Isoprenoids	α-Terpineol	B	y = 154.66x + 0.8754	1216	43
Sulcatone	Isoprenoids	Sulcatone	A	y = 645.87x + 0.9362	1330	43
Camphor	Isoprenoids	Limonene	B	y = 3.7759x + 0.0009	1501	95
Linalool	Isoprenoids	Linalool	A	y = 3.4965x + 0.0045	1520	71
α-Ionene	Isoprenoids	α-ionone	A	y = 753.46x + 0.0117	1559	159
Hotrienol	Isoprenoids	β-Myrcene	A	y = 1.2298x + 9E-05	1588	71
Levomenthol	Isoprenoids	α-Terpineol	B	y = 154.66x + 0.8754	1642	71
α-Terpineol	Isoprenoids	α-Terpineol	A	y = 154.66x + 0.8754	1683	59
β-Damascenone	Isoprenoids	β-Damascenone	A	y = 65.15x + 1.447	1803	69
cis-Geranylacetone	Isoprenoids	Neral	B	y = 2583.1x + 121.51	1832	69
3,3,5-Trimethylcyclohexene	Others	Limonene	B	y = 3.7759x + 0.0009	1572	109
3,4,4-Trimethyl-2-cyclopenten-1-one	Others	Limonene	B	y = 3.7759x + 0.0009	1573	109
γ-Caprolactone	Others	Limonene	B	y = 3.7759x + 0.0009	1693	85

^a^ Identification of the compounds, ‘A’ means identified by mass spectrum and RI agree with standards, ‘B’ means tentatively identified by mass spectrum agrees with the mass spectral database and RI agrees with literature. ^b^ Retention indices on DB-wax column. ^c^ Quantitative ion.

## Data Availability

Data are contained within the article.

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
