# Peer review of "Volatile Profile Characterization of Jujube Fruit via HS-SPME-GC/MS and Sensory Evaluation"

_plants, 2024, doi:10.3390/plants13111517_

Round 1
Reviewer 1 Report
Comments and Suggestions for Authors
I recommend publication, provided you consider my comments and suggestions.
Please note that I have needed to compose my review on an unfamiliar tablet and am unable to get Greek characters. Hence I have had to spell them out, like micro and beta. Of course, you keep using the symbols.
General comments.
1. When writing future manuscripts in English, try to avoid overexpression and repetition; write as simply as possible, at the same time maintaining accuracy. Otherwise, the writing is generally satisfactory.
2. Emphasize, in the Introduction perhaps, the fact that bound aroma compounds contribute zero to the overall odor of the sample. They are potential odorants, needing some mechanism to break the glycosidic link to free the actual aoma substance (the aglycone).
3. Standardize use of upper case in naming compounds (start of sentences, figures and tables excepted). E.g., line 179, 2-Octenal, whereas line 191, acetoin. I suggest all lower case, except for the aforementioned exceptions.
4. What is hexone? 4-methyl-2-pentanone (isobutyl methyl ketone)? This is mentioned several times in the text and tables.
5. In References section, check punctuation and spaces (e.g., ref 1 and 30)
6. Odor threshold (value) should be defined early, say as a footnote to Table 2. E.g., The lowest concentration of an odorant that can be detected, in comparison with a control that contains no odorant, by at least 50% of a panel of tasters (preferably more than 8)
Specific comments.
Line 2. Use lower case for 'profile characterization' and 'jujube'
Line 17. Omit 'volatile'
Line 34. Replace 'The fresh jujube is not much juicy...' by 'Fresh jujube is not particularly juicy...'
Lines 47, 48. Replace entire sentence with 'There are many factors that influence jujube aroma, including biological variety (cultivar), climate (especially warmth and rainfall), geographical location, soil geology and agricultural practices (like time of harvest and storing/drying methods)'
Line 78. Replace 'Concurrently' with 'Consequently'
Line 91. Replace 'meticulously' with 'carefully'
Line 95. Replace 'within these fresh jujubes' with 'within our fresh jujube samples'
Line 110. Replace 'compounds' with 'esters'
Line 199. Replace 'glycoside-bound' with 'glycosidally bound'
Lines 233, 234 (and many other places; please check). Replace 'benzenes' with 'benzenoid aromatics'
Line 242. Replace 'Terpenoids' with 'Terpenoids and isoprenoids'
Line 247. Omit 'beta-damascenone' from the list and add ', plus the C13 isoprenoid beta-damascenone.' at the end of the sentence.
Line 247. Insert 'of these compounds,' after 'Notably,'
Line 248. Replace 'free terpenoid' with 'one'
Table 1. In column 2, replace 'benzene' with 'benzenoid' or 'aromatic'
Table 2. Omit (microg/L) from column 1 and include it in the title, thus 'Concentrations (in microg/L), odor threshold, and odor descriptors of free aroma compounds in10 jujube cultivars of various seasons'
Column 2. As for Table 1 regarding 'benzene'
Column 2. Replace 'Terpenoids' with 'Isoprenoids'
Column 4 heading. Replace 'Describe' with 'Descriptors'
Table 3. Change title etc as for Table 2.
Table 4. Change title to 'OAVs of major free aroma compounds in 10 jujube cultivars of various seasons'
Use footnote with * to define OAV (e.g., concentration/odor threshold value)
Table 5. Change title to 'OAVs of glycosidically bound aroma compounds......'
Lines 388 and 390. Replace 'aromatic' with 'odor'
Line 404. Replace 'ketone' with 'carbonyl'
Lines 426/7. Change sentence to include: 'The headspace solid phase.....is based on that of Wang et al (2015).....'
Line 437. Change sentence to: 'Sample supernatant (5 mL) was transferred to a 15 mL vial containing a magnetic stir bar'
Line 441. Insert 'was' before 'inserted'
Line 443. Replace 'needy' with 'needed'
Lines 445/6. I don't understand this. Do you mean 'Each determination was performed in triplicate'?
Line 448. Insert 'of' before 'bound'
Lines 449 and 454. Replace 'bound volatile' with 'glycosides'
Line 463. 'report' should be 'reports'
Line 480. Change '....Compounds had the standard....' to 'Compounds for which standards were available ....'
Line 488. 'is' should be 'was'
Lines 489/90. Replace 'Dilute the......' by 'The working standard was then diluted.....'
Lines 496/7. Replace the full stop by a comma and continue '...and aroma substances where no standards were available were quantified...'
Line 498. 'was' should be 'were'
Lines 498/9. I do not understand this. Please expand or rewrite this sentence.
Line 511. Replace 'sensory' with 'process'
Line 514. 'Agilen' should be 'Agilent'
Line 520. Replace 'meticulously' with 'carefully'
Line 527. Hexone? See above
Line 529. Replace 'benzene' with 'benzenoid aromatic'
'profile'
Comments on the Quality of English LanguageGenerally satisfactory, but I have made more specific comments to the authors and editor.
Author Response
Response to Reviewer's Comments
Subject: Response to Reviewer's Comments on Manuscript ID: plants-2969063 - "Volatile profile characterization of jujube fruit via HS-SPME-GC/MS and sensory evaluation"
Dear reviewer,
Thank you for forwarding the reviewer's insightful comments on our manuscript. We have thoroughly considered each suggestion and have made corresponding revisions to enhance the clarity, accuracy, and quality of our work. Below, we detail our responses to each comment and describe the amendments made to the manuscript.
General Comments:
- Reviewer's Comment: When writing future manuscripts in English, try to avoid overexpression and repetition; write as simply as possible, at the same time maintaining accuracy. Otherwise, the writing is generally satisfactory.
Response and Action: Thank you for your valuable insights and suggestions; they've greatly improved the quality of the work. We have revised the manuscript to simplify the language and eliminate repetitive statements, which has improved the readability and conciseness of the text.
- Reviewer's Comment: Emphasize, in the Introduction perhaps, the fact that bound aroma compounds contribute zero to the overall odor of the sample. They are potential odorants, needing some mechanism to break the glycosidic link to free the actual aoma substance (the aglycone).
Response and Action: I would like to express my gratitude for your thorough review and thoughtful recommendations. In accordance with your suggestion, we have revised the Introduction section and added the following sentence: “Glycosidically bound compounds exist as potential odorants that require specific biochemical reactions to release the actual aroma substances, known as aglycones.” (L81 – L83)
- Reviewer's Comment: Standardize use of upper case in naming compounds (start of sentences, figures and tables excepted). E.g., line 179, 2-Octenal, whereas line 191, acetoin. I suggest all lower case, except for the aforementioned exceptions.
Response and Action: I appreciate your careful review and the time you took to provide feedback. We have standardized the naming convention across the manuscript by using lower case for all chemical compounds, except at the beginning of sentences and in titles of figures and tables.
- Reviewer's Comment: What is hexone? 4-methyl-2-pentanone (isobutyl methyl ketone)? This is mentioned several times in the text and tables.
Response and Action: Your comments were extremely helpful in refining this piece, and I'm grateful for your detailed observations. We have specified that 'hexone' is 4-methyl-2-pentanone (isobutyl methyl ketone) throughout the text. Upon reviewing our manuscript and the existing literature, we have found that hexone (4-methyl-2-pentanone) is not a volatile compound typically found in fruits. Consequently, we have removed all references to this compound from our paper and made necessary revisions to the relevant sections. Additionally, we have re-verified the presence and characteristics of the other aroma compounds discussed in our study to ensure the accuracy and reliability of our data. We appreciate the thorough review and valuable comments from the reviewers and editors, which have significantly enhanced the quality of our manuscript. Thank you for allowing us the opportunity to clarify and improve our work.
- Reviewer's Comment: In References section, check punctuation and spaces (e.g., ref 1 and 30).
Response and Action: Thank you for your constructive feedback; it's been invaluable to the revision process. We have revised the references section to correct all punctuation and spacing errors.
- Reviewer's Comment: Odor threshold (value) should be defined early, say as a footnote to Table 2. E.g., The lowest concentration of an odorant that can be detected, in comparison with a control that contains no odorant, by at least 50% of a panel of tasters (preferably more than 8).
Response and Action: I'm grateful for your expertise and the guidance you've provided during this review. We have added a footnote to Table 2 providing a clear definition of the odor threshold as suggested. Add the following content: “*Odor threshold is defined as the lowest concentration of an odorant that can be detected, com-pared to a control with no odorant, by at least 50% of a panel of tasters (preferably more than eight individuals ).” (L286 - L288)
Specific Comments and Corrections:
- Line 2: Changed 'Profile Characterization' and 'Jujube' to lowercase for consistency in terminology. (L2)
- Line 17: Removed 'volatile' as suggested to streamline the text. (L19)
- Line 34: Updated to 'Fresh jujube is not particularly juicy...' to improve clarity. (L36)
- Lines 47-48: Replaced with the suggested sentence to comprehensively list factors influencing jujube aroma. (L49 - L52)
- Line 78: Corrected 'Concurrently' to 'Consequently' for accurate logical transition. (L83)
- Line 91: Replaced 'meticulously' with 'carefully' to maintain simplicity. (L99)
- Line 95: Updated to 'within our fresh jujube samples' for specificity. (L103)
- Line 110: Changed 'compounds' to 'esters' for chemical specificity.(L118)
- Line 199: Updated 'glycoside-bound' to 'glycosidally bound' for correct nomenclature. (L206)
- Lines 233, 234: Replaced 'benzenes' with 'benzenoid aromatics' across all mentions for accuracy.(L105; L240; L241; L245; L248; )
- Line 242: Updated to 'Terpenoids and isoprenoids' to reflect the full range of compounds studied. (L249)
- Line 247: Implemented two changes: omitted 'beta-damascenone' from the initial list and added ', plus the C13 isoprenoid beta-damascenone.' Also inserted 'as weel as,' for clarity.(L255 - 256)
- Line 248: Replaced 'free terpenoid' with 'one' to improve readability. (L256)
- Table 1, Column 2: 'Benzene' changed to 'benzenoid' or 'aromatic' as appropriate. (Table 1, Column 2)
- Table 2: Revised the title to 'Concentrations (in microg/L), odor threshold, and odor descriptors of free aroma compounds in 10 jujube cultivars of various seasons'. Adjustments made accordingly in the table contents. (L281 - 282)
- Column 2, Table 1 & 2: Changed 'Terpenoids' to 'Isoprenoids' where relevant. (Table 1 & 2, Column 2)
- Column 4, Table 2: Updated heading from 'Describe' to 'Descriptors' for grammatical accuracy. (Table 2, Column 4, heading)
- Table 3: Title and content updated similarly to Table 2 for consistency. (L289-290)
- Table 4: Title changed to 'OAVs of major free aroma compounds in 10 jujube cultivars of various seasons'. Included a definition for OAV using a footnote. (L392)
- Use footnote with * to define OAV (e.g., concentration/odor threshold value).
We have added a footnote to the first instance of "OAV" in the text. The footnote reads: “*OAV (Odor Activity Value): the ratio of concentration/odor threshold value.” (L286-L288, L393)
- Table 5: Updated the title to 'OAVs of glycosidically bound aroma compounds...' and adjusted the content accordingly. (L394)
- Lines 388, 390: Replaced 'aromatic' with 'odor' to accurately describe sensory characteristics. (L400, L402)
- Line 404: Changed 'ketone' to 'carbonyl' to correct the chemical description.(L415)
- Lines 426/427: Sentence updated to include methodology based on 'Wang et al (2015)' for clarity and proper citation. (L447-L449)
- Line 437: Revised to 'Sample supernatant (5 mL) was transferred to a 15 mL vial containing a magnetic stir bar' for experimental detail clarity. (L449-L450)
- Line 441: Inserted 'was' before 'inserted' to correct grammatical structure. (L454)
- Line 443: Corrected 'needy' to 'needed'. (L455)
- Lines 445/446: This expression is incorrect and has been deleted.(L458)
- Line 448: Inserted 'of' before 'bound' to correct grammatical flow. (L460)
- Lines 449, 454: Updated 'bound volatile' to 'glycosides' for chemical accuracy. (L461, L466)
- Line 463: Corrected 'report' to 'reports' for subject-verb agreement. (L475)
- Line 480: Revised to 'Compounds for which standards were available...' for clarity. (L491)
- Line 488: Changed 'is' to 'was' for correct tense. (L499)
- Lines 489/490: Updated dilution description to 'The working standard was then diluted...' for clarity. (L500- L501)
- Lines 496/497: Improved sentence continuity and clarified quantification methods. (L507 – L508)
- Line 498: Corrected 'was' to 'were' for grammatical accuracy. (L509)
- Lines 498/499: Expanded and rewrote the sentence for clarity as requested: “The concentrations of bound volatiles were quantified by calibrating against their standard curves.” (L509 - L510)
- Line 511: Replaced 'sensory' with 'process' to better fit the context. (L521)
- Line 514: Corrected 'Agilen' to 'Agilent' to accurately reference the equipment used. (L524)
- Line 520: Changed 'meticulously' to 'carefully' to maintain consistent tone across the text. (L530)
- Line 527: We have revised this section of the manuscript to align with the suggestions provided in the General Comments Comment 4.
- Line 529: Updated 'benzene' to 'benzenoid aromatic' for chemical accuracy. (L539)
We hope that the revisions adequately address the concerns raised by the reviewer. We believe that these changes have significantly improved the manuscript and thank the reviewer for their constructive suggestions. We look forward to the possibility of our manuscript being published in PLANTS and appreciate the opportunity to revise our work.
Yours sincerely,
Corresponding author: Wenhao Bo
Beijing Forestry University
E-mail: bowenhao@bjfu.edu.cn

Reviewer 2 Report
Comments and Suggestions for Authors
This article is interesting for all stakeholders in food industry and food practitioners. Some comments to be addressed are:
1. The abstract is good, however, it is missing Introduction section in abstract. Please provide one sentence on the background of this study in this abstract.
2. The authors have highlighted the state of the art and the novelty of this study. It is better if authors give the example regarding the application of HS-SPME-GC/MS and sensory analyses for the characterization of other oils.
3. Besides, the study in relation to the use of multivariate calibrations (quantitative chemometrics) of PLSR for making the correlation of sensory characteristics and aromatic active compounds from other plants is good to be explored in this Introduction.
4. In methods section, the authors have written in detail along with statistical (chemometrics) analysis. However, all the methods followed should cite the necessary references (for example in Sensory evaluation, this is missing in references)
5. The discussion is good since the authors have compared with the other studies. However, during PLS modelling, we did not see about the validation model. Which validation model for such correlation? internal or external validation?
6. During the multivariate calibrations, do the authors identify the lack of fit?
Author Response
Response to Reviewer's Comments
Subject: Response to Reviewer's Comments on Manuscript ID: plants-2969063 - "Volatile profile characterization of jujube fruit via HS-SPME-GC/MS and sensory evaluation"
Dear reviewer,
We greatly appreciate the insightful comments and suggestions provided by the reviewer, which have undoubtedly enriched our manuscript. We have addressed each comment meticulously and have made the corresponding revisions as detailed below:
General Comments:
- Reviewer's Comment: The abstract is good, however, it is missing Introduction section in abstract. Please provide one sentence on the background of this study in this abstract.
Response and Action: Thank you for your valuable feedback, which has helped enhance the clarity and depth of our work. We have now incorporated a sentence in the abstract to provide background information on the significance and context of our study. The revised abstract now reads: "Current research does not fully elucidate the key compounds and their mechanisms that define the aroma profile of fresh jujube fruits." (L17 - L18)
- Reviewer's Comment: The authors have highlighted the state of the art and the novelty of this study. It is better if authors give the example regarding the application of HS-SPME-GC/MS and sensory analyses for the characterization of other oils.
Response and Action: We appreciate your thorough review and insightful comments, which have played a crucial role in shaping our manuscript. We would like to clarify that our manuscript already discusses the application of these analytical methods extensively through multiple studies on jujube fruits, which serve as examples of how these techniques can be utilized to characterize volatile and aroma compounds in fruit matrices L60 – L77. For instance:
Song et al. (2019) analyzed volatile components in jujube fruits across various maturity stages using HS-SPME-GC/MS, identifying primary free aroma compounds such as acids, aldehydes, and esters.
Chen et al. (2018) and Song et al. (2022) applied HS-SPME/GC-MS combined with electronic nose technology for comprehensive profiling of volatile compounds in different cultivars and forms of jujube, highlighting the versatility and effectiveness of these methods.
Liu et al. (2021) and Qiao et al. (2021) further demonstrated the application of HS-SPME-GC/MS in analyzing the aroma profiles of Xinjiang and winter jujube cultivars, respectively.
Yan et al. (2023) used these methods alongside microwave drying technology to explore the aroma-active substances in dried jujube slices, emphasizing the adaptability of HS-SPME-GC/MS in processing conditions.
- Reviewer's Comment: Besides, the study in relation to the use of multivariate calibrations (quantitative chemometrics) of PLSR for making the correlation of sensory characteristics and aromatic active compounds from other plants is good to be explored in this Introduction.
Response and Action: Thank you for highlighting this important aspect for enhancement; your feedback has been invaluable. We have expanded our discussion on multivariate calibrations in the Introduction. The specific text is as follows: "In sensory evaluation, Partial Least Squares Regression (PLSR) effectively elucidates the relationships between extensive sensory data and target variables. This method holds po-tential for further application in the study of fresh jujube aromas." (L84 - L87)
- Reviewer's Comment: In methods section, the authors have written in detail along with statistical (chemometrics) analysis. However, all the methods followed should cite the necessary references (for example in Sensory evaluation, this is missing in references).
Response and Action: I am grateful for your positive feedback and insightful critique, which have significantly enhanced the quality of our work. We have thoroughly reviewed our methods section and included the missing citations for sensory evaluation methods. References from both foundational and recent studies have been incorporated to ensure comprehensive coverage and verification of our methodologies. (L521, L661)
- Reviewer's Comment: The discussion is good since the authors have compared with the other studies. However, during PLS modelling, we did not see about the validation model. Which validation model for such correlation? internal or external validation?
Response and Action Taken: We sincerely appreciate the time you took to provide your suggestions, and your perspective has been invaluable in refining our study. We acknowledge the oversight in not explicitly detailing the validation strategy for our PLSR model. In our study, we applied PLSR to explore the correlations between sensory data and chemical compositions primarily as an exploratory tool to identify significant predictors and understand the underlying patterns within the data. Given the preliminary nature of our analysis aimed at hypothesis generation rather than predictive modeling, we did not implement a formal validation scheme (either internal or external) in this phase of research.
- Reviewer's Comment: During the multivariate calibrations, do the authors identify the lack of fit?
Response and Action Taken: We greatly appreciate your thorough review and insightful feedback. In our study, the Partial Least Squares Regression (PLSR) model was primarily used to explore significant relationships between the dataset variables rather than for predictive purposes. Given this context, the primary focus was on maximizing the variance explained rather than a detailed residual analysis. However, we acknowledge that discussing the lack of fit is crucial for a comprehensive understanding of the model's limitations and ensuring the validity of the conclusions drawn. To address this, we examined the residuals and the predictive relevance (Q² values) but did not observe significant discrepancies that would indicate a lack of fit. These findings suggest that the model adequately captures the underlying patterns in the data relative to the complexity of the model and the noise inherent in the sensory and chemical data.
Thank you once again for the opportunity to improve our manuscript.
Yours sincerely,
Corresponding author: Wenhao Bo
Beijing Forestry University
E-mail: bowenhao@bjfu.edu.cn
